# Deep Ensemble of Slime Mold Algorithm and Arithmetic Optimization Algorithm for Global Optimization

Rong Zheng [1,*][iD], Heming Jia [1,*][iD], Laith Abualigah [2,3,4], Qingxin Liu [5][iD] and Shuang Wang [1]

1   School of Information Engineering, Sanming University, Sanming 365004, China; wang_shuang@fjsmu.edu.cn
2   Research and Innovation Department, Skyline University College, Sharjah 1797, United Arab Emirates; Aligah.2020@gmail.com
3   Faculty of Computer Sciences and Informatics, Amman Arab University, Amman 11953, Jordan
4   School of Computer Science, Universiti Sains Malaysia, Gelugor 11800, Malaysia
5   School of Computer Science and Technology, Hainan University, Haikou 570228, China; qxliu@hainanu.edu.cn
*   Correspondence: zhengr@fjsmu.edu.cn (R.Z.); jiaheming@fjsmu.edu.cn (H.J.)

**Abstract:** In this paper, a new hybrid algorithm based on two meta-heuristic algorithms is presented to improve the optimization capability of original algorithms. This hybrid algorithm is realized by the deep ensemble of two new proposed meta-heuristic methods, i.e., slime mold algorithm (SMA) and arithmetic optimization algorithm (AOA), called DESMAOA. To be specific, a preliminary hybrid method was applied to obtain the improved SMA, called SMAOA. Then, two strategies that were extracted from the SMA and AOA, respectively, were embedded into SMAOA to boost the optimizing speed and accuracy of the solution. The optimization performance of the proposed DESMAOA was analyzed by using 23 classical benchmark functions. Firstly, the impacts of different components are discussed. Then, the exploitation and exploration capabilities, convergence behaviors, and performances are evaluated in detail. Cases at different dimensions also were investigated. Compared with the SMA, AOA, and another five well-known optimization algorithms, the results showed that the proposed method can outperform other optimization algorithms with high superiority. Finally, three classical engineering design problems were employed to illustrate the capability of the proposed algorithm for solving the practical problems. The results also indicate that the DESMAOA has very promising performance when solving these problems.

**Keywords:** slime mold algorithm; arithmetic optimization algorithm; meta-heuristics algorithm; global optimization; engineering design problem

## 1. Introduction

Nowadays, optimization problems exist in various scenarios, for instance, the engineering design problems. The objective of these optimization problems is to find the extreme values with determined constraint conditions. Then, commonly, the cost is reduced as much as possible. To tackle these problems, researchers have proposed many optimization algorithms [1–4]. Generally speaking, traditional optimization algorithms, such as gradient-based methods, are susceptible to the initial positions and have difficulties to deal with the non-convex problems that may contain a mass of local optimums. In practice, when we are faced with complex constraint conditions in the real world, it is more essential to obtain the optimal solutions within limited time and cost. At this point, the important thing is not to find the theoretical optimal result but to obtain as good an approximate solution as possible under restricted conditions. For this purpose, many stochastic optimizers have been developed and employed to solve complex optimization problems. As its name implies, the random operator is the main feature for a stochastic optimizer, which allows the algorithms to avoid the stagnation and search the whole search region for global optimization result.

The meta-heuristic algorithms (MAs) have shown very powerful capability in the fields of computational sciences. In general, MAs have four types according to the sources of inspiration, namely, physics-inspired (PI), evolution-inspired (EI), swarm-inspired (SI), and human-inspired (HI). Some representative algorithms are shown below:

- Physics-inspired: multi-verse optimizer (MVO) [5], gravitational search algorithm (GSA) [6], thermal exchange optimization (TEO) [7], heat transfer relation-based optimization algorithm (HTOA) [8].
- Evolution-inspired: genetic algorithm (GA) [9], differential evolution (DE) [10], evolutionary programming (EP) [11].
- Swarm-inspired: particle swarm optimization (PSO) [12], emperor penguin optimizer (EPO) [13], Aquila optimizer (AO) [14], remora optimization algorithm (ROA) [4], marine predators algorithm (MPA) [15].
- Human-inspired: teaching–learning-based optimization (TLBO) [16], social group optimization (SGO) [17], β-hill climbing (βHC) [18], coronavirus optimization algorithm (COA) [19].

For a meta-heuristic algorithm, one important thing is to balance of the global search and local search [20]. It is known that the search agents are first randomly generated within the search spaces. Then, positions of these search agents are updated according to the formulas in the algorithm. In the early stage, drastic exploration in the search space should be performed as much as possible in the early stage. Then, in the later phase, more local exploitation should be conducted to improve the accuracy of obtained optimal solution. Although hundreds of MAs have been proposed for the optimization problems, there is still need for new algorithms to solve these optimization problems. According to the No-Free-Lunch (NFL) theory [21], on one optimization algorithm can solve all the optimization problems. Generally speaking, it is common that MAs suffer from local optimum stagnation and poor convergence speed as a result of poor optimization ability. Thus, it is very important to develop new optimization algorithms or improve existing MAs by taking some effective measures. Up until now, there have been three primary methods for the improvements of the existing algorithms, which are listed in Table 1.

The slime mold algorithm (SMA) [33] and arithmetic optimization algorithm (AOA) [34] are two newly proposed MAs, and both have the merits of simplicity, efficiency, and flexibility. The SMA has good population diversity and stable performance when solving optimization problems. However, it gets stuck in local optima sometimes for the limited global search capability. On the contrary, the AOA has powerful exploration capability by using the arithmetic operators. However, the performance of AOA is not stable because of the poor population diversity. Therefore, the SMA and AOA are considered to be hybridized together in this paper for solving the global optimization problems. To evaluate the performance of proposed algorithm, we employed 23 classical benchmark functions and 4 constrained engineering design problems. The main contributions of this works are as follows:

1. Hybridizing the slime mold algorithm (SMA) [33] and arithmetic optimization algorithm (AOA) [34] named SMAOA to improve the exploration capability of original SMA.
2. Applying the random contraction strategy (RCS), which is inspired from SMA to help the SMAOA jump out from local optimum.
3. Applying the subtraction and addition strategy (SAS), which is extracted from AOA to enhance the exploitation ability of SMAOA.
4. When the RCS and SAS were applied on SMAOA, the DESMAOA was finally obtained. By comparing seven well-known optimization algorithms, we identified the proposed DESMAOA to be powerful according to the experimental results.

**Table 1.** A summary of methods for improving the optimization algorithms developed in the literature.

| Name of Method | Representative Algorithm | Description |
|---|---|---|
| Hybridize two or more algorithms | Hybrid sperm swarm optimization and gravitational search algorithm (HSSOGSA) [22] | The capability of exploitation in SSO and the capability of exploration in GSA are combined for better performance. |
| | Imperialist competitive Harris hawks optimization (ICHHO) [23] | The exploration of ICA is utilized to improve the HHO for global optimization. |
| | Hybrid particle swarm and spotted hyena optimizer (HPSSHO) [24] | Particle swarm algorithm is used to improve the hunting strategy of spotted hyena optimizer. |
| Add one or more strategies onto an algorithm | Sine-cosine and spotted hyena-based chimp optimization algorithm (SSC) [25] | Sine-cosine functions and attacking strategy of SHO are embedded in ChoA for better exploration and exploitation. |
| | Representative-based grey wolf optimizer (R-GWO) [26] | A search strategy named representative-based hunting (RH) is utilized to improve the exploration and diversity of the population |
| | Reinforced salp swarm algorithm (CMSRSSSA) [27] | An ensemble/composite mutation strategy (CMS) is applied to boost the exploitation and exploration speed of SSA, while restart strategy (RS) is used to get away from local optimum. |
| | Boosting quantum rotation gate embedded slime mold algorithm (WQSMA) [28] | The quantum rotation gate mechanism and the operation from water cycle are applied to balance the exploration and exploitation inclinations. |
| | Enhanced salp swarm algorithm (ESSA) [29] | Orthogonal learning, quadratic interpolation, and generalized oppositional learning are embedded into SSA to boost the global exploration and local exploitation. |
| Hybridize two or more algorithms that are further improved by one or more strategies | Whale optimization with seagull algorithm (WSOA) [30] | WOA's contraction surrounding mechanism and SOA's spiral attack behavior work together, and then levy flight strategy is employed on the search process of SOA. |
| | Chaotic sine-cosine firefly (CSCF) algorithm [31] | Chaotic form of SCA and FA are integrated together to improve the convergence speed and efficiency. |
| | Hybrid grasshopper optimization algorithm with bat algorithm (BGOA) [32] | In BGOA, Levy fight, local search part of BA, and random strategy are introduced into basic GOA. |

The rest of this paper is organized as follows: The basics of SMA and AOA are described in Section 2. Then, the hybrid method is presented in Section 3, including two strategies that are obtained from these two algorithms. In Section 4, a series of experimental tests are conducted to evaluate the performance of proposed DESMAOA. In Section 5, three engineering design problems are employed to assess the applicability of proposed algorithm in practice. Finally, Section 6 concludes this paper and provides some directions for meaningful future research.

## 2. Preliminaries

### 2.1. Slime Mold Algorithm (SMA)

The slime mold algorithm (SMA) is a recent meta-heuristic algorithm proposed by Li et al. in 2020 [33]. The basic idea of SMA is based on the foraging behavior of slime mold, which have different feedback characteristics according to the food quality. Three special behaviors of the slime mold are mathematical formulated in the SMA, i.e., approaching food, wrapping food, and finally grabbling food. First, the process of approaching food can be expressed as

$$Xi(t+1) = \begin{cases} Xb(t) + vb \cdot (W \cdot XA(t) - XB(t)), & r1 < p \\ vc \cdot Xi(t), & r1 \geq p \end{cases} \tag{1}$$

where $t$ is the number of current iteration, $X_i(t+1)$ is the newly generated position, $X_b(t)$ denotes the best position found by slime mold in iteration $t$, $X_A(t)$ and $X_B(t)$ are two random positions selected from the population of slime mold, and $r_1$ is a random value in [0, 1].

$vb$ and $vc$ are the coefficients that simulate the oscillation and contraction mode of slime mold, respectively, and $vc$ is designed to linearly decrease from one to zero during the iterations. The range of $vb$ is from $-a$ to $a$, and the computational formula of $a$ is

$$a = \text{arctanh}(1 - \frac{t}{T}) \tag{2}$$

where $T$ is the maximum number of iterations.

According to Equations (1) and (2), it can be seen that as the number of iterations increases, the slime mold will wrap the food.

*W* is a very important factor that indicates the weight of slime mold, and it is calculated as follows:

$$W(SmellIndex(i)) = \begin{cases} 1 + rand \cdot \log(\frac{bF - S(i)}{bF - wF} + 1), & i \leq N/2 \\ 1 - rand \cdot \log(\frac{bF - S(i)}{bF - wF} + 1), & i > N/2 \end{cases} \tag{3}$$

$$SmellIndex(i) = sort(S(i)) \tag{4}$$

where *rand* means a random value between 0 and 1; *bF* and *wF* are the best and worst fitness values, respectively, obtained by far; $S(i)$ is the fitness value of *i*th slime mold; *N* is the popsize of the population; and *SmellIndex* is a ranking of fitness values for individuals in the population.

In Equation (1), it is also worth noting that *p* is the probability of determining the update location for slime mold, which is related to the fitness values of slime mold and food and can be calculated as follows:

$$p = \tanh|S(i) - DF| \tag{5}$$

where *DF* denotes the best fitness obtained by population.

Finally, when the slime mold has found the food (i.e., grabble food), it still has a certain chance (*z*) to search other new food, which is formulated as

$$X(t + 1) = rand \cdot (UB - LB) + LB, \; r_2 < z \tag{6}$$

where *UB* and *LB* are the upper boundary and lower boundary, respectively, and $r_2$ implies a random value in the region [0, 1].

In general, *z* should be very small; thus, it is set to 0.03 in SMA. Finally, the pseudo-code of SMA is given in Algorithm 1.

---

**Algorithm 1.** Pseudo-code of SMA

---

Initialize the parameters popsize (*N*) and maximum iterations (*T*)
Initialize the positions of all slime mold $X_i$ ($i$ = 1, 2, ... , *N*)
**While** ($t \leq T$)
Calculate the fitness of all slime mold
Update *bestFitness*, $X_b$
Calculate the weight *W* by Equation (3) and (4)
**For** each search agent
**If** $r_2 < z$
Update position by Equation (6)
**Else**
Update *p*, *vb*, and *vc*
Update position by Equation (1)
**End if**
**End for**
$t = t + 1$
**End While**
**Return** *bestFitness*, $X_b$

---

### 2.2. Arithmetic Optimization Algorithm (AOA)

Arithmetic optimization algorithm (AOA) is a very new meta-heuristic method proposed by Abualigah and others in 2021 [34]. The main inspiration of this algorithm is to combine the four traditional arithmetic operators in mathematics, i.e., multiplication (*M*), division (*D*), subtraction (*S*), and addition (*A*). Similar to sine-cosine algorithm (SCA) [35], AOA also has a very simple structure and low computation complexity. Considering the

*M* and *D* operators can produce large steps in the iterations, *M* and *D* are hence mainly conducted in the exploration phase. The expression is as follows:

$$Xi(t+1) = \begin{cases} Xb(t)/(MOP + eps) \cdot ((UB - LB)\mu + LB), \ rand < 0.5 \\ Xb(t) \cdot MOP \cdot ((UB - LB)\mu + LB), \ rand \geq 0.5 \end{cases} \tag{7}$$

where *eps* is a very small positive number, and $\mu$ is a constant coefficient (0.499) that is carefully designed for this algorithm.

*MOP* is non-linearly decreased from 1 to 0 during the iterations, and the expression is as follows:

$$MOP = 1 - (\frac{t}{T})^{1/\alpha} \tag{8}$$

where $\alpha$ is a constant value, which is set to 5 according to the AOA.

From Equation (7), it can be seen that both *M* and *D* operators can generate very stochastic positions for the search agent on the basis of the best position. By contrast, *S* and *A* operators are applied to emphasize the local exploitation that will generate smaller steps in the search space. The mathematical expression is defined as

$$Xi(t+1) = \begin{cases} Xb(t) - MOP \cdot ((UB - LB)\mu + LB), \ rand < 0.5 \\ Xb(t) + MOP \cdot ((UB - LB)\mu + LB), \ rand \geq 0.5 \end{cases} \tag{9}$$

There is no doubt that the importance of balance between exploration and exploitation for an optimization algorithm. In AOA, the parameter *MOA* is utilized to switch the exploration and exploitation over the course of iterations, which is expressed as

$$MOA(t) = Min + t(\frac{Max - Min}{T}) \tag{10}$$

where *Min* and *Max* are constant values.

According to Equation (10), *MOA* increases from *Min* to *Max*. Thus, in the early phase, search agent has more chance to perform exploration in the search space, while in the later stage, search agent will be more likely to conduct search near the best position. The pseudo-code of AOA is shown in Algorithm 2.

---

**Algorithm 2.** Pseudo-code of AOA

---

Initialize the parameters popsize (*N*) and maximum iterations (*T*)
Initialize the positions of all search agents $X_i$ (*i* = 1, 2, . . . , *N*)
Set the parameters $\alpha$, $\mu$, *Min*, and *Max*
**While** ($t \leq T$)
Calculate the fitness of all search agents
Update *bestFitness*, $X_b$
Calculate the *MOP* by Equation (8)
Calculate the *MOA* by Equation (10)
**For** each search agent
**If** *rand* > *MOA*
Update position by Equation (7)
**Else**
Update position by Equation (9)
**End if**
**End for**
$t = t + 1$
**End While**
**Return** *bestFitness*, $X_b$

---

### 3. The Proposed Hybridized Algorithm (DESMAOA)

It is well known that MAs have the merits of concision, flexibility, and especially utility. Hence, many scholars are working on developing new meta-heuristic-based approaches for optimization problems. However, several optimization algorithms such as slime mold algorithm and arithmetic optimization algorithm still have some drawbacks. For instance, when dealing with complex optimization problems, SMA tends to drop into local best, and also converges slowly. Similarly, AOA only utilizes the information of best position in the population, which may suffer the problem of low precision. Therefore, this paper aimed to develop a new hybridization algorithm composed of SMA and AOA for better optimization performance.

In this paper, the SMA and AOA are firstly integrated to form a hybridized style named SMAOA. Then, the preliminary hybrid algorithm is further enhanced by adding two strategies. One is the random contraction strategy (RCS), which is an improved version of contraction formula in SMA. The other is the subtraction and addition strategy (SAS), which is extracted from the local search in AOA. Finally, the deep ensemble of SMA and AOA is accomplished, and the hybridized algorithm (i.e., DESMAOA) is obtained. The detailed implement of proposed algorithm is delineated in the following.

#### 3.1. The Hybridization of SMA and AOA

In SMA, the contraction formula (see Equations (1) and (2)) is utilized to help slime mold jump out of local minima, which will tend to zero in the later iterations. Thus, it will not play the role of global exploration. On the other hand, the multiplication and division methods in AOA display a powerful capability in global exploration. Thus, the formulas of multiplication and division (see in Equation (7)) are considered to replace the contraction equation. Therefore, the hybrid algorithm SMAOA will perform good global search in the whole stage. To be specific, for the search agent that is close to best position, the multiplication and division operators will make it more likely to search other spaces.

#### 3.2. Random Contraction Strategy (RCS)

In this work, we present the RCS on the basis of the mathematical formula of contraction mode in SMA, which is applied to expand exploration space and avoid local optimum. The coefficient $vc$ is replaced by a random value lying between $-1$ and $1$. The position update formula is calculated as follows:

$$Vi2(t+1) = (2\,rand - 1)Xi(t) \tag{11}$$

From Equation (11), we should note that the generated position of RCS is within the range $[-\mid X_i(t)\mid, \mid X_i(t)\mid]$ with uniform distribution, which adds more flexibility for the search agents in the proposed algorithm. Note that the generated position of RCS is taken as a candidate solution.

#### 3.3. Subtraction and Addition Strategy (SAS)

The other strategy proposed here is the SAS, which is also the exploitation method of AOA. According to the AOA, SAS can be performed locally and increase the accuracy of solutions effectively. It is worth mentioning here that the SAS is conducted behind the RCS.

In the same way, the position generated by SAS is treated as a candidate solution, and if a better position is found, then it will be adopted.

#### 3.4. The Deep Ensemble of SMA and AOA

As mentioned above, the SMA and AOA are hybridized together firstly to achieve the SMAOA. Then, two strategies are introduced in the SMAOA, namely, random contraction strategy and subtraction and addition strategy. In order to perform a better balance effect between exploration and exploitation, we utilize a parameter, $b$, that is related with

iterations to represent the probability of conducting the strategies. Its computational formula is given below:

$$b = 1 - \frac{t}{T} \tag{12}$$

The pseudo-code of DESMAOA is shown in Algorithm 3. Moreover, the flowchart of proposed method is shown in Figure 1.

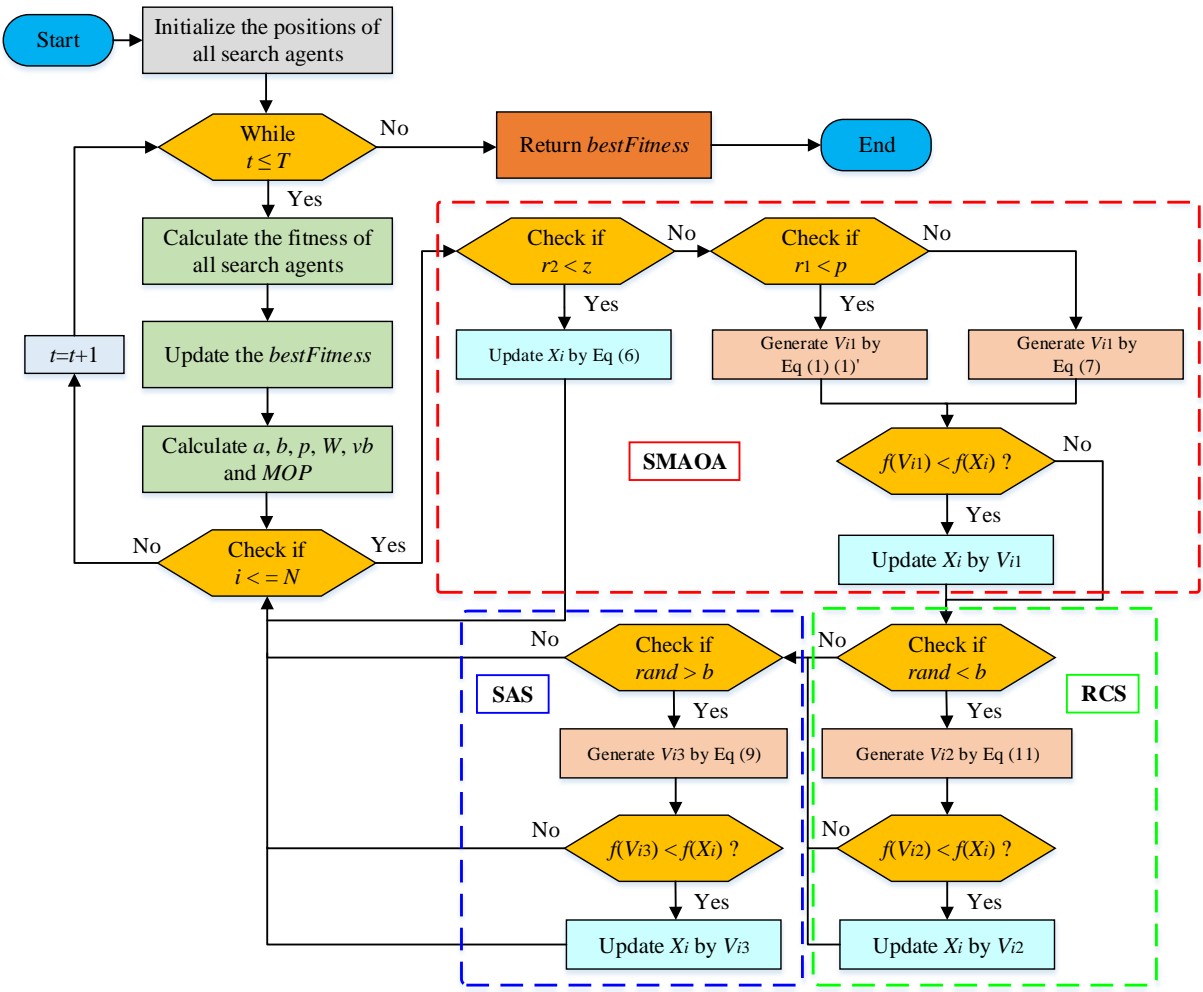

**Figure 1.** Flowchart of the proposed DESMAOA.

### 3.5. The Computational Complexity of DESMAOA

The computational complexity of DESMAOA depends on the population size (*N*), dimension size (*D*), and maximum iterations (*T*). First, the computational complexity of initialization is $O(N \times D)$. Then, in the iterations, the computational complexity of calculating the fitness values of all search agents is $O(N)$. The computational complexity of sorting is $O(N \times \log N)$. Moreover, the computational complexity of updating the positions of search agents in SMAOA is $O(N \times D)$. Considering the worst cases, the computational complexity of RCS and SAS is $O(2N \times D)$. In summary, the final computational complexity of the DESMAOA is $O(N \times D + T \times N(1 + \log N + 3D))$.

---

**Algorithm 3.** Pseudo-code of DESMAOA

---

Initialize the parameters popsize ($N$) and maximum iterations ($T$)
Initialize the positions of all search agents $X_i$ ($i = 1, 2, \ldots , N$)
Set the parameters $\alpha$, $\mu$, *Min*, and *Max*
**While** ($t \leq T$)
Calculate the fitness of all search agents
Update *bestFitness*, $X_b$
Calculate $a$, $b$, $p$, and $W$ by Equation (2)–(5)
Calculate the *MOP* by Equation (8)
Update *vb*
**For** each search agent
**If** $r_2 < z$
Update position by Equation (6)
**Else**
**If** $r_1 < p$
Update position $V_{i1}$ by Equation (1) (1)′
**Else**
Update position $V_{i1}$ by Equation (7)
**End if**
**If** $f(V_{i1}) < f(X_i)$
$X_i = V_{i1}$
**End if**
**If** $rand < b$
Apply RCS and generate candidate position $V_{i2}$ by Equation (11)
**If** $f(V_{i2}) < f(X_i)$
$X_i = V_{i2}$
**End if**
**End if**
**If** $rand > b$
Apply SAS and generate candidate position $V_{i3}$ by Equation (9)
**If** $f(V_{i3}) < f(X_i)$
$X_i = V_{i3}$
**End if**
**End if**
**End if**
**End for**
$t = t + 1$
**End While**
**Return** *bestFitness*, $X_b$

---

## 4. Experimental Results and Discussions

In this section, we provide the results of a series of comparative experiments that were conducted by using 23 classical benchmark functions and 10 IEEE CEC2021 single objective optimization functions to evaluate the performance of proposed DESMAOA [36,37]. Table 2 lists the detailed parameter values of these test functions. It can be seen that these classical test functions included unimodal functions (F1–F7), multimodal functions (F8–F13), and also fixed-dimension multimodal functions (F14–F23). Moreover, the CEC2021 test functions contained four types of functions: unimodal function, basic functions, hybrid functions, and composition functions. The unimodal functions are suitable for testing the exploitation capability of algorithms, while the other types of test functions that contain a large number of local minimas can reveal the exploration capability and stability of algorithms.

In the experiments of test functions, the impacts of two applied strategies were firstly analyzed by using the classical test functions. Then, the test results of DESMAOA in classical test functions were compared with seven well-known algorithms. Multiple aspects of the analysis including exploitation capability, exploration capability, qualitative analysis, and convergence behavior are described. Moreover, the results of CEC2021 test functions were also analyzed to investigate the performance of proposed algorithm.

<div align="center">**Table 2.** Benchmark function properties (*D* indicates dimension).</div>

| Function Type | Function | Dimension | Range | Theoretical Optimization Value |
|---|---|---|---|---|
| Unimodal test functions | F1 | 30, 50, 200, 1000 | $[-100, 100]$ | 0 |
| | F2 | 30, 50, 200, 1000 | $[-10, 10]$ | 0 |
| | F3 | 30, 50, 200, 1000 | $[-100, 100]$ | 0 |
| | F4 | 30, 50, 200, 1000 | $[-100, 100]$ | 0 |
| | F5 | 30, 50, 200, 1000 | $[-30, 30]$ | 0 |
| | F6 | 30, 50, 200, 1000 | $[-100, 100]$ | 0 |
| | F7 | 30, 50, 200, 1000 | $[-1.28, 1.28]$ | 0 |
| Multimodal test functions | F8 | 30, 50, 200, 1000 | $[-500, 500]$ | $-418.9829 \times D$ |
| | F9 | 30, 50, 200, 1000 | $[-5.12, 5.12]$ | 0 |
| | F10 | 30, 50, 200, 1000 | $[-32, 32]$ | 0 |
| | F11 | 30, 50, 200, 1000 | $[-600, 600]$ | 0 |
| | F12 | 30, 50, 200, 1000 | $[-50, 50]$ | 0 |
| | F13 | 30, 50, 200, 1000 | $[-50, 50]$ | 0 |
| Fixed-dimension multimodal test functions | F14 | 2 | $[-65, 65]$ | 0.998004 |
| | F15 | 4 | $[-5, 5]$ | 0.0003075 |
| | F16 | 2 | $[-5, 5]$ | $-1.03163$ |
| | F17 | 2 | $[-5, 5]$ | 0.398 |
| | F18 | 2 | $[-2, 2]$ | 3 |
| | F19 | 3 | $[-1, 2]$ | $-3.8628$ |
| | F20 | 6 | $[0, 1]$ | $-3.3220$ |
| | F21 | 4 | $[0, 10]$ | $-10.1532$ |
| | F22 | 4 | $[0, 10]$ | $-10.4028$ |
| | F23 | 4 | $[0, 10]$ | $-10.5363$ |
| CEC2021 unimodal test functions | CEC_01 | 10 | $[-100, 100]$ | 100 |
| CEC2021 basic test functions | CEC_02 | 10 | $[-100, 100]$ | 1100 |
| | CEC_03 | 10 | $[-100, 100]$ | 700 |
| | CEC_04 | 10 | $[-100, 100]$ | 1900 |
| CEC2021 hybrid test functions | CEC_05 | 10 | $[-100, 100]$ | 1700 |
| | CEC_06 | 10 | $[-100, 100]$ | 1600 |
| | CEC_07 | 10 | $[-100, 100]$ | 2100 |
| CEC2021 composition test functions | CEC_08 | 10 | $[-100, 100]$ | 2200 |
| | CEC_09 | 10 | $[-100, 100]$ | 2400 |
| | CEC_10 | 10 | $[-100, 100]$ | 2500 |

*4.1. Impacts of Components*

The impacts of different versions are investigated in this section. SMA showed very outstanding performance in optimization problems. However, it still had the problems of premature convergence and local optima. According to the works of Abualigah [34], AOA shows powerful global exploration and local exploitation capability. Hence, we first hybridized the SMA and AOA to obtain the SMAOA. Then, in order to help the search agent jump out of local optima, we integrated the RCS into SMAOA. Moreover, SAS was introduced into SMAOA to improve the local search capability. Different combinations between SMAOA and two strategies are listed below:

- SMAOA;
- SMAOA combined with RCS (SMAOA1);
- SMAOA combined with SAS (SMAOA2);
- SMAOA combined with RCS and SAS (DESMAOA).

For impartial comparison, the number of iterations and population size for all tests were set as 500 and 30, respectively. Moreover, we conducted independent tests 30 times for each algorithm. The averages and standard deviations were utilized for analysis and comparison between these algorithms. The results are listed in Table 3. Note that the dimension of F1–F13 was set to 30.

**Table 3.** Comparison of the SMAOA, SMAOA1, SMAOA2, and DESMAOA.

| Function | SMAOA | | SMAOA1 | | SMAOA2 | | DESMAOA | |
|---|---|---|---|---|---|---|---|---|
| | Mean | Std | Mean | Std | Mean | Std | Mean | Std |
| F1 | $0.00 \times 10^0$ | $0.00 \times 10^0$ | $0.00 \times 10^0$ | $0.00 \times 10^0$ | $0.00 \times 10^0$ | $0.00 \times 10^0$ | $0.00 \times 10^0$ | $0.00 \times 10^0$ |
| F2 | $0.00 \times 10^0$ | $0.00 \times 10^0$ | $0.00 \times 10^0$ | $0.00 \times 10^0$ | $0.00 \times 10^0$ | $0.00 \times 10^0$ | $0.00 \times 10^0$ | $0.00 \times 10^0$ |
| F3 | $0.00 \times 10^0$ | $0.00 \times 10^0$ | $0.00 \times 10^0$ | $0.00 \times 10^0$ | $0.00 \times 10^0$ | $0.00 \times 10^0$ | $0.00 \times 10^0$ | $0.00 \times 10^0$ |
| F4 | $0.00 \times 10^0$ | $0.00 \times 10^0$ | $0.00 \times 10^0$ | $0.00 \times 10^0$ | $0.00 \times 10^0$ | $0.00 \times 10^0$ | $0.00 \times 10^0$ | $0.00 \times 10^0$ |
| F5 | $2.82 \times 10^0$ | $7.00 \times 10^0$ | $5.85 \times 10^{-1}$ | $1.07 \times 10^0$ | $2.46 \times 10^{-1}$ | $1.33 \times 10^0$ | $1.17 \times 10^{-3}$ | $1.55 \times 10^{-3}$ |
| F6 | $2.44 \times 10^{-2}$ | $2.34 \times 10^{-2}$ | $7.77 \times 10^{-3}$ | $1.21 \times 10^{-2}$ | $5.80 \times 10^{-6}$ | $1.92 \times 10^{-6}$ | $4.95 \times 10^{-6}$ | $2.01 \times 10^{-6}$ |
| F7 | $1.16 \times 10^{-4}$ | $9.72 \times 10^{-5}$ | $5.67 \times 10^{-5}$ | $5.61 \times 10^{-5}$ | $6.72 \times 10^{-5}$ | $6.77 \times 10^{-5}$ | $4.27 \times 10^{-5}$ | $4.76 \times 10^{-5}$ |
| F8 | $-12{,}569.2361$ | $1.89 \times 10^{-1}$ | $-12{,}569.3229$ | $1.38 \times 10^{-1}$ | $-12{,}569.4866$ | $4.96 \times 10^{-6}$ | $-12{,}569.4866$ | $4.11 \times 10^{-6}$ |
| F9 | $0.00 \times 10^0$ | $0.00 \times 10^0$ | $0.00 \times 10^0$ | $0.00 \times 10^0$ | $0.00 \times 10^0$ | $0.00 \times 10^0$ | $0.00 \times 10^0$ | $0.00 \times 10^0$ |
| F10 | $8.8818 \times 10^{-16}$ | $0.00 \times 10^0$ | $8.8818 \times 10^{-16}$ | $0.00 \times 10^0$ | $8.8818 \times 10^{-16}$ | $0.00 \times 10^0$ | $8.8818 \times 10^{-16}$ | $0.00 \times 10^0$ |
| F11 | $2.13 \times 10^{-1}$ | $2.92 \times 10^{-1}$ | $0.00 \times 10^0$ | $0.00 \times 10^0$ | $8.85 \times 10^{-3}$ | $2.30 \times 10^{-2}$ | $0.00 \times 10^0$ | $0.00 \times 10^0$ |
| F12 | $2.63 \times 10^{-3}$ | $4.65 \times 10^{-3}$ | $5.08 \times 10^{-5}$ | $8.33 \times 10^{-5}$ | $4.84 \times 10^{-8}$ | $7.54 \times 10^{-8}$ | $1.09 \times 10^{-7}$ | $1.59 \times 10^{-7}$ |
| F13 | $1.70 \times 10^{-2}$ | $3.67 \times 10^{-2}$ | $8.82 \times 10^{-4}$ | $1.15 \times 10^{-3}$ | $2.50 \times 10^{-3}$ | $6.04 \times 10^{-3}$ | $5.91 \times 10^{-7}$ | $1.06 \times 10^{-6}$ |
| F14 | $9.98 \times 10^{-1}$ | $2.60 \times 10^{-11}$ | $9.98 \times 10^{-1}$ | $9.58 \times 10^{-12}$ | $9.98 \times 10^{-1}$ | $9.91 \times 10^{-16}$ | $9.98 \times 10^{-1}$ | $8.05 \times 10^{-16}$ |
| F15 | $4.16 \times 10^{-4}$ | $1.56 \times 10^{-4}$ | $3.63 \times 10^{-4}$ | $9.61 \times 10^{-5}$ | $4.07 \times 10^{-4}$ | $1.90 \times 10^{-4}$ | $3.34 \times 10^{-4}$ | $8.63 \times 10^{-5}$ |
| F16 | $-1.0316 \times 10^0$ | $3.97 \times 10^{-8}$ | $-1.0316 \times 10^0$ | $9.46 \times 10^{-8}$ | $-1.0316 \times 10^0$ | $1.67 \times 10^{-11}$ | $-1.0316 \times 10^0$ | $1.87 \times 10^{-11}$ |
| F17 | $3.9789 \times 10^{-1}$ | $6.82 \times 10^{-7}$ | $3.9789 \times 10^{-1}$ | $3.97 \times 10^{-7}$ | $3.9789 \times 10^{-1}$ | $5.63 \times 10^{-12}$ | $3.9789 \times 10^{-1}$ | $5.94 \times 10^{-12}$ |
| F18 | $3.00 \times 10^0$ | $2.79 \times 10^{-9}$ | $3.00 \times 10^0$ | $6.69 \times 10^{-10}$ | $3.00 \times 10^0$ | $8.56 \times 10^{-11}$ | $3.00 \times 10^0$ | $9.42 \times 10^{-11}$ |
| F19 | $-3.8627 \times 10^0$ | $4.32 \times 10^{-5}$ | $-3.8628 \times 10^0$ | $4.68 \times 10^{-5}$ | $-3.8628 \times 10^0$ | $5.61 \times 10^{-5}$ | $-3.8627 \times 10^0$ | $7.91 \times 10^{-5}$ |
| F20 | $-3.25 \times 10^0$ | $5.98 \times 10^{-2}$ | $-3.2859 \times 10^0$ | $5.59 \times 10^{-2}$ | $-3.2583 \times 10^0$ | $6.06 \times 10^{-2}$ | $-3.286 \times 10^0$ | $5.59 \times 10^{-2}$ |
| F21 | $-1.01528 \times 10^1$ | $4.26 \times 10^{-4}$ | $-1.01529 \times 10^1$ | $3.67 \times 10^{-4}$ | $-1.01531 \times 10^1$ | $9.07 \times 10^{-5}$ | $-1.01531 \times 10^1$ | $1.42 \times 10^{-4}$ |
| F22 | $-1.04023 \times 10^1$ | $4.61 \times 10^{-4}$ | $-1.04025 \times 10^1$ | $4.47 \times 10^{-4}$ | $-1.04028 \times 10^1$ | $8.31 \times 10^{-5}$ | $-1.04028 \times 10^1$ | $7.38 \times 10^{-5}$ |
| F23 | $-1.0536 \times 10^1$ | $3.47 \times 10^{-4}$ | $-1.05362 \times 10^1$ | $2.47 \times 10^{-4}$ | $-1.05363 \times 10^1$ | $9.44 \times 10^{-5}$ | $-1.05363 \times 10^1$ | $8.26 \times 10^{-5}$ |

From Table 3, it can be seen that these four improved algorithms could obtain the same optimal fitness in F1–F4, F9, F10, F14, and F16–F18. In particular, the theoretical optimization values were obtained in F1–F4, F9, and F18. Compared to SMAOA, SMAOA1, and SMAOA2, DESMAOA won in F5–F8, F11, F13, F15, and F20–F24. This demonstrates that the significant effect with the combination of RCS and SAS. In addition, it is worth mentioning here that the results of DESMAOA in F12 and F19 were very close to the best ones. From the values of standard deviations, it was also shown that DESMAOA had good stability and strong robustness in solving these test functions.

### 4.2. The Classical Benchmark Functions

This section outlines the 23 classical test functions that were employed for experiments. The performance of DESMAOA was compared with two newly proposed algorithms (SMA and AOA) and another five very famous optimization algorithms (GWO [38], WOA [39], SSA [40], MVO [5], and PSO [12]). Table 4 lists the main parameter values used in each algorithm. Note that the parameter values used in DESMAOA were the same as those used in two original algorithms. Therefore, the stable performance could be guaranteed to some extent for the proposed algorithm. In addition, the test conditions were the same as previously for equal comparison.

**Table 4.** Parameter values for the optimization algorithms.

| Algorithm | Parameter Settings |
|---|---|
| DESMAOA | $z = 0.03$; $\alpha = 5$; $\mu = 0.499$ |
| SMA [33] | $z = 0.03$ |
| AOA [34] | $\alpha = 5$; $\mu = 0.499$; $Min = 0.2$; $Max = 1$ |
| GWO [38] | $a = [2, 0]$ |
| WOA [39] | $a_1 = [2, 0]$; $a_2 = [-2, -1]$; $b = 1$ |
| SSA [40] | $c_1 \in [0, 1]$; $c_2 \in [0, 1]$ |
| MVO [5] | $WEP \in [0.2, 1]$; $TDR \in [0, 1]$; $r_1, r_2, r_3 \in [0, 1]$ |
| PSO [12] | $c_1 = 2$; $c_2 = 2$; $W \in [0.2, 0.9]$; $vMax = 6$ |

4.2.1. Exploration and Exploitation Capability Analysis

Table 5 lists the experimental results of these algorithms. It was shown that the performance of DESMAOA is not only better than the original SMA and AOA but also superior to other comparative algorithms on 20 out of 23 benchmark functions. In F1–F5, F7–F15, F22, and F23, DESMAOA had the lowest average values and stand deviations. This reveals that DESMAOA possesses very good stability and also can find the optimal solution. It is worth noting that the proposed algorithm can obtain the theoretical optimization values in test functions F1–F4, F9, F11, and F18. In F6, F19, and F20, the results of DESMAOA were very close to the best ones. Therefore, these results demonstrated the remarkable effect of the proposed hybrid method. With the help of RCS, the proposed algorithm can jump out of the local minima and obtain the global optimal solution. In the meantime, high precision results could be obtained by using SAS.

In addition, the Wilcoxon signed-rank test was utilized to confirm the statistical superiority of DESMAOA [41], which revealing the statistical differences between two algorithms. The results are given in Table 6. On the basis of these results and the results in Table 5, DESMAOA outperformed SMA for 15 benchmark functions (except F1, F3, F7, F9, F10, F11, F15, and F20) and AOA for 20 benchmark functions (except F7, F15, and F17). Moreover, DESMAOA was found to be better than other comparative algorithms in most of the functions. Furthermore, the results of test functions were also evaluated using the Friedman ranking test [42], which can reveal the overall performance ranking of the comparative algorithms to the test functions. As can be seen from Figure 2, the proposed DESMAOA achieved the first rank among these algorithms. In summary, DESMAOA had excellent optimization performance that was significantly better than SMA and AOA.

**Table 5.** The result statistics of benchmark functions for the DESMAOA and competitor algorithms.

| Function | Metric | DESMAOA | SMA | AOA | GWO | WOA | SSA | MVO | PSO |
|---|---|---|---|---|---|---|---|---|---|
| F1 | Mean | $0.00 \times 10^0$ | $9.93 \times 10^{-302}$ | $5.37 \times 10^{-6}$ | $7.21 \times 10^{-28}$ | $2.42 \times 10^{-73}$ | $3.96 \times 10^{-7}$ | $1.34 \times 10^0$ | $1.76 \times 10^{-4}$ |
| | Std | $0.00 \times 10^0$ | $0.00 \times 10^0$ | $2.14 \times 10^{-6}$ | $1.17 \times 10^{-27}$ | $8.81 \times 10^{-73}$ | $9.50 \times 10^{-7}$ | $5.38 \times 10^{-1}$ | $1.82 \times 10^{-4}$ |
| F2 | Mean | $0.00 \times 10^0$ | $5.05 \times 10^{-138}$ | $1.74 \times 10^{-3}$ | $8.26 \times 10^{-17}$ | $8.81 \times 10^{-52}$ | $2.07 \times 10^0$ | $2.20 \times 10^{-1}$ | $7.05 \times 10^0$ |
| | Std | $0.00 \times 10^0$ | $2.77 \times 10^{-137}$ | $2.08 \times 10^{-3}$ | $6.54 \times 10^{-17}$ | $2.46 \times 10^{-51}$ | $1.33 \times 10^0$ | $7.31 \times 10^0$ | $7.01 \times 10^0$ |
| F3 | Mean | $0.00 \times 10^0$ | $5.43 \times 10^{-323}$ | $1.24 \times 10^{-3}$ | $1.55 \times 10^{-5}$ | $4.41 \times 10^4$ | $1.66 \times 10^3$ | $2.04 \times 10^2$ | $7.93 \times 10^1$ |
| | Std | $0.00 \times 10^0$ | $0.00 \times 10^0$ | $8.14 \times 10^{-4}$ | $3.50 \times 10^{-5}$ | $1.08 \times 10^4$ | $9.24 \times 10^2$ | $6.63 \times 10^1$ | $2.57 \times 10^1$ |
| F4 | Mean | $0.00 \times 10^0$ | $7.56 \times 10^{-154}$ | $1.53 \times 10^{-2}$ | $8.03 \times 10^{-7}$ | $4.68 \times 10^1$ | $1.15 \times 10^1$ | $2.16 \times 10^0$ | $1.12 \times 10^0$ |
| | Std | $0.00 \times 10^0$ | $4.14 \times 10^{-153}$ | $1.06 \times 10^{-2}$ | $6.71 \times 10^{-7}$ | $2.77 \times 10^1$ | $4.04 \times 10^0$ | $8.66 \times 10^{-1}$ | $2.40 \times 10^{-1}$ |
| F5 | Mean | $1.17 \times 10^{-3}$ | $8.56 \times 10^0$ | $2.79 \times 10^1$ | $2.71 \times 10^1$ | $2.82 \times 10^1$ | $2.90 \times 10^2$ | $7.89 \times 10^2$ | $8.16 \times 10^1$ |
| | Std | $1.55 \times 10^{-3}$ | $1.21 \times 10^1$ | $3.01 \times 10^{-1}$ | $8.49 \times 10^{-1}$ | $4.97 \times 10^{-1}$ | $4.77 \times 10^2$ | $8.74 \times 10^2$ | $7.03 \times 10^1$ |
| F6 | Mean | $4.95 \times 10^{-6}$ | $5.74 \times 10^{-3}$ | $3.06 \times 10^0$ | $7.58 \times 10^{-1}$ | $3.72 \times 10^{-1}$ | $1.78 \times 10^{-7}$ | $1.34 \times 10^0$ | $1.37 \times 10^{-4}$ |
| | Std | $2.01 \times 10^{-6}$ | $3.38 \times 10^{-3}$ | $2.69 \times 10^{-1}$ | $4.94 \times 10^{-1}$ | $2.18 \times 10^{-1}$ | $1.51 \times 10^{-7}$ | $3.43 \times 10^{-1}$ | $1.65 \times 10^{-4}$ |
| F7 | Mean | $4.27 \times 10^{-5}$ | $1.24 \times 10^{-4}$ | $6.74 \times 10^{-5}$ | $1.69 \times 10^{-3}$ | $3.15 \times 10^{-3}$ | $1.73 \times 10^{-1}$ | $3.21 \times 10^{-2}$ | $2.55 \times 10^0$ |
| | Std | $4.76 \times 10^{-5}$ | $1.07 \times 10^{-4}$ | $7.11 \times 10^{-5}$ | $8.95 \times 10^{-4}$ | $3.61 \times 10^{-3}$ | $5.61 \times 10^{-2}$ | $1.32 \times 10^{-2}$ | $4.54 \times 10^0$ |
| F8 | Mean | $-12{,}569.4866$ | $-12{,}569.1799$ | $-5.48 \times 10^3$ | $-6.01 \times 10^3$ | $-1.06 \times 10^4$ | $-7.47 \times 10^3$ | $-7.55 \times 10^3$ | $-4.69 \times 10^3$ |
| | Std | $4.11 \times 10^{-6}$ | $2.66 \times 10^{-1}$ | $3.69 \times 10^2$ | $6.42 \times 10^2$ | $1.69 \times 10^3$ | $8.76 \times 10^2$ | $6.27 \times 10^2$ | $1.21 \times 10^3$ |
| F9 | Mean | $0.00 \times 10^0$ | $0.00 \times 10^0$ | $1.66 \times 10^{-6}$ | $2.26 \times 10^0$ | $3.79 \times 10^{-15}$ | $5.53 \times 10^1$ | $1.20 \times 10^2$ | $1.02 \times 10^2$ |
| | Std | $0.00 \times 10^0$ | $0.00 \times 10^0$ | $1.27 \times 10^{-6}$ | $3.27 \times 10^0$ | $2.08 \times 10^{-14}$ | $1.83 \times 10^1$ | $3.29 \times 10^1$ | $3.19 \times 10^1$ |
| F10 | Mean | $8.8818 \times 10^{-16}$ | $8.8818 \times 10^{-16}$ | $4.36 \times 10^{-4}$ | $1.01 \times 10^{-13}$ | $3.85 \times 10^{-15}$ | $2.56 \times 10^0$ | $2.03 \times 10^0$ | $1.69 \times 10^{-2}$ |
| | Std | $0.00 \times 10^0$ | $0.00 \times 10^0$ | $1.62 \times 10^{-4}$ | $1.81 \times 10^{-14}$ | $2.10 \times 10^{-15}$ | $6.94 \times 10^{-1}$ | $5.47 \times 10^{-1}$ | $1.30 \times 10^{-2}$ |
| F11 | Mean | $0.00 \times 10^0$ | $0.00 \times 10^0$ | $8.42 \times 10^{-4}$ | $6.19 \times 10^{-3}$ | $1.68 \times 10^{-2}$ | $1.88 \times 10^{-2}$ | $8.60 \times 10^{-1}$ | $4.39 \times 10^{-3}$ |
| | Std | $0.00 \times 10^0$ | $0.00 \times 10^0$ | $3.12 \times 10^{-3}$ | $8.92 \times 10^{-3}$ | $6.38 \times 10^{-2}$ | $1.46 \times 10^{-2}$ | $8.21 \times 10^{-2}$ | $6.85 \times 10^{-3}$ |
| F12 | Mean | $1.09 \times 10^{-7}$ | $5.81 \times 10^{-3}$ | $7.44 \times 10^{-1}$ | $4.42 \times 10^{-2}$ | $2.81 \times 10^{-2}$ | $7.54 \times 10^0$ | $2.43 \times 10^0$ | $2.07 \times 10^0$ |
| | Std | $1.59 \times 10^{-7}$ | $6.50 \times 10^{-3}$ | $3.03 \times 10^{-2}$ | $1.86 \times 10^{-2}$ | $2.18 \times 10^{-2}$ | $3.43 \times 10^0$ | $1.39 \times 10^0$ | $4.22 \times 10^{-2}$ |
| F13 | Mean | $5.91 \times 10^{-7}$ | $6.35 \times 10^{-3}$ | $2.96 \times 10^0$ | $6.94 \times 10^{-1}$ | $6.36 \times 10^{-1}$ | $1.38 \times 10^1$ | $1.96 \times 10^{-1}$ | $5.55 \times 10^{-3}$ |
| | Std | $1.06 \times 10^{-6}$ | $7.05 \times 10^{-3}$ | $1.03 \times 10^{-2}$ | $2.45 \times 10^{-1}$ | $3.53 \times 10^{-1}$ | $1.10 \times 10^1$ | $1.26 \times 10^{-1}$ | $9.01 \times 10^{-3}$ |
| F14 | Mean | $9.98 \times 10^{-1}$ | $9.98 \times 10^{-1}$ | $9.87 \times 10^{-1}$ | $4.16 \times 10^0$ | $2.54 \times 10^0$ | $1.36 \times 10^0$ | $9.98 \times 10^{-1}$ | $2.97 \times 10^0$ |
| | Std | $8.05 \times 10^{-16}$ | $3.93 \times 10^{-13}$ | $3.89 \times 10^0$ | $4.28 \times 10^0$ | $2.91 \times 10^0$ | $8.82 \times 10^{-1}$ | $4.31 \times 10^{-11}$ | $2.55 \times 10^0$ |
| F15 | Mean | $3.34 \times 10^{-4}$ | $4.84 \times 10^{-4}$ | $8.39 \times 10^{-3}$ | $3.15 \times 10^{-3}$ | $8.25 \times 10^{-4}$ | $2.91 \times 10^{-3}$ | $5.24 \times 10^{-3}$ | $7.22 \times 10^{-3}$ |
| | Std | $8.63 \times 10^{-5}$ | $2.15 \times 10^{-4}$ | $1.29 \times 10^{-2}$ | $6.88 \times 10^{-3}$ | $5.40 \times 10^{-4}$ | $5.93 \times 10^{-3}$ | $1.26 \times 10^{-2}$ | $9.03 \times 10^{-3}$ |
| F16 | Mean | $-1.0316 \times 10^0$ | $-1.0316 \times 10^0$ | $-1.0316 \times 10^0$ | $-1.0316 \times 10^0$ | $-1.0316 \times 10^0$ | $-1.0316 \times 10^0$ | $-1.0316 \times 10^0$ | $-1.0316 \times 10^0$ |
| | Std | $1.87 \times 10^{-11}$ | $8.36 \times 10^{-10}$ | $2.28 \times 10^{-11}$ | $3.13 \times 10^{-8}$ | $1.68 \times 10^{-9}$ | $3.89 \times 10^{-14}$ | $4.19 \times 10^{-7}$ | $6.25 \times 10^{-16}$ |
| F17 | Mean | $3.9789 \times 10^{-1}$ | $3.9789 \times 10^{-1}$ | $4.0217 \times 10^{-1}$ | $3.9789 \times 10^{-1}$ | $3.9789 \times 10^{-1}$ | $3.9789 \times 10^{-1}$ | $3.9789 \times 10^{-1}$ | $3.9789 \times 10^{-1}$ |
| | Std | $5.94 \times 10^{-12}$ | $5.40 \times 10^{-9}$ | $1.52 \times 10^{-2}$ | $8.10 \times 10^{-7}$ | $6.71 \times 10^{-6}$ | $7.99 \times 10^{-15}$ | $1.27 \times 10^{-7}$ | $0.00 \times 10^0$ |
| F18 | Mean | $3.0000 \times 10^0$ | $3.0000 \times 10^0$ | $4.8000 \times 10^0$ | $5.7000 \times 10^0$ | $3.0001 \times 10^0$ | $3.0000 \times 10^0$ | $3.0000 \times 10^0$ | $3.0000 \times 10^0$ |
| | Std | $9.42 \times 10^{-11}$ | $1.17 \times 10^{-9}$ | $6.85 \times 10^0$ | $1.48 \times 10^1$ | $8.14 \times 10^{-5}$ | $2.13 \times 10^{-13}$ | $3.49 \times 10^{-6}$ | $1.79 \times 10^{-15}$ |
| F19 | Mean | $-3.8627 \times 10^0$ | $-3.8628 \times 10^0$ | $-3.8627 \times 10^0$ | $-3.8605 \times 10^0$ | $-3.8572 \times 10^0$ | $-3.8628 \times 10^0$ | $-3.8628 \times 10^0$ | $-3.8628 \times 10^0$ |
| | Std | $7.91 \times 10^{-5}$ | $1.58 \times 10^{-7}$ | $2.62 \times 10^{-4}$ | $4.08 \times 10^{-3}$ | $1.02 \times 10^{-2}$ | $1.17 \times 10^{-12}$ | $7.73 \times 10^{-6}$ | $2.58 \times 10^{-15}$ |
| F20 | Mean | $-3.286 \times 10^0$ | $-3.2503 \times 10^0$ | $-3.2942 \times 10^0$ | $-3.2339 \times 10^0$ | $-3.2225 \times 10^0$ | $-3.2255 \times 10^0$ | $-3.2454 \times 10^0$ | $-3.2402 \times 10^0$ |
| | Std | $5.59 \times 10^{-2}$ | $5.95 \times 10^{-2}$ | $5.12 \times 10^{-2}$ | $7.30 \times 10^{-2}$ | $1.10 \times 10^{-1}$ | $5.45 \times 10^{-2}$ | $5.93 \times 10^{-2}$ | $8.13 \times 10^{-2}$ |
| F21 | Mean | $-1.01531 \times 10^1$ | $-1.01531 \times 10^1$ | $-7.8781 \times 10^0$ | $-8.8066 \times 10^0$ | $-8.4438 \times 10^0$ | $-6.9755 \times 10^0$ | $-7.048 \times 10^0$ | $-6.3883 \times 10^0$ |
| | Std | $1.42 \times 10^{-4}$ | $1.05 \times 10^{-4}$ | $2.68 \times 10^0$ | $2.54 \times 10^0$ | $2.44 \times 10^0$ | $3.35 \times 10^0$ | $3.28 \times 10^0$ | $3.27 \times 10^0$ |
| F22 | Mean | $-1.04028 \times 10^1$ | $-1.04028 \times 10^1$ | $-7.2814 \times 10^0$ | $-1.02239 \times 10^1$ | $-7.0271 \times 10^0$ | $-8.938 \times 10^0$ | $-9.0327 \times 10^0$ | $-8.71250 \times 10^0$ |
| | Std | $7.38 \times 10^{-5}$ | $1.82 \times 10^{-4}$ | $3.52 \times 10^0$ | $9.70 \times 10^{-1}$ | $3.08 \times 10^0$ | $2.99 \times 10^0$ | $2.83 \times 10^0$ | $2.91 \times 10^0$ |
| F23 | Mean | $-1.05363 \times 10^1$ | $-1.05363 \times 10^1$ | $-6.6743 \times 10^0$ | $-1.05349 \times 10^1$ | $-7.7815 \times 10^0$ | $-8.1138 \times 10^0$ | $-8.5201 \times 10^0$ | $-9.1233 \times 10^0$ |
| | Std | $8.26 \times 10^{-5}$ | $9.71 \times 10^{-5}$ | $3.31 \times 10^0$ | $8.48 \times 10^{-4}$ | $3.28 \times 10^0$ | $3.51 \times 10^0$ | $3.20 \times 10^0$ | $2.93 \times 10^0$ |

**Table 6.** *p*-values of the Wilcoxon signed-rank test between DESMAOA and other competitor algorithms.

| Function | DESMAOA vs. SMA | DESMAOA vs. AOA | DESMAOA vs. GWO | DESMAOA vs. WOA | DESMAOA vs. SSA | DESMAOA vs. MVO | DESMAOA vs. PSO |
|---|---|---|---|---|---|---|---|
| F1 | $1.00 \times 10^{0}$ | $6.10 \times 10^{-5}$ | $6.10 \times 10^{-5}$ | $6.10 \times 10^{-5}$ | $6.10 \times 10^{-5}$ | $6.10 \times 10^{-5}$ | $6.10 \times 10^{-5}$ |
| F2 | $6.10 \times 10^{-5}$ | $6.10 \times 10^{-5}$ | $6.10 \times 10^{-5}$ | $6.10 \times 10^{-5}$ | $6.10 \times 10^{-5}$ | $6.10 \times 10^{-5}$ | $6.10 \times 10^{-5}$ |
| F3 | $1.00 \times 10^{0}$ | $6.10 \times 10^{-5}$ | $6.10 \times 10^{-5}$ | $6.10 \times 10^{-5}$ | $6.10 \times 10^{-5}$ | $6.10 \times 10^{-5}$ | $6.10 \times 10^{-5}$ |
| F4 | $6.10 \times 10^{-5}$ | $6.10 \times 10^{-5}$ | $6.10 \times 10^{-5}$ | $6.10 \times 10^{-5}$ | $6.10 \times 10^{-5}$ | $6.10 \times 10^{-5}$ | $6.10 \times 10^{-5}$ |
| F5 | $6.10 \times 10^{-5}$ | $6.10 \times 10^{-5}$ | $6.10 \times 10^{-5}$ | $6.10 \times 10^{-5}$ | $6.10 \times 10^{-5}$ | $6.10 \times 10^{-5}$ | $6.10 \times 10^{-5}$ |
| F6 | $1.22 \times 10^{-4}$ | $6.10 \times 10^{-5}$ | $6.10 \times 10^{-5}$ | $6.10 \times 10^{-5}$ | $6.10 \times 10^{-5}$ | $6.10 \times 10^{-5}$ | $8.54 \times 10^{-4}$ |
| F7 | $2.52 \times 10^{-1}$ | $1.88 \times 10^{-1}$ | $6.10 \times 10^{-5}$ | $6.10 \times 10^{-5}$ | $6.10 \times 10^{-5}$ | $6.10 \times 10^{-5}$ | $6.10 \times 10^{-5}$ |
| F8 | $6.10 \times 10^{-5}$ | $6.10 \times 10^{-5}$ | $6.10 \times 10^{-5}$ | $6.10 \times 10^{-5}$ | $6.10 \times 10^{-5}$ | $6.10 \times 10^{-5}$ | $6.10 \times 10^{-5}$ |
| F9 | $1.00 \times 10^{0}$ | $1.22 \times 10^{-4}$ | $6.10 \times 10^{-5}$ | $1.00 \times 10^{0}$ | $6.10 \times 10^{-5}$ | $6.10 \times 10^{-5}$ | $6.10 \times 10^{-5}$ |
| F10 | $1.00 \times 10^{0}$ | $6.10 \times 10^{-5}$ | $6.10 \times 10^{-5}$ | $9.77 \times 10^{-4}$ | $6.10 \times 10^{-5}$ | $6.10 \times 10^{-5}$ | $6.10 \times 10^{-5}$ |
| F11 | $1.00 \times 10^{0}$ | $6.10 \times 10^{-5}$ | $2.50 \times 10^{-1}$ | $1.00 \times 10^{0}$ | $6.10 \times 10^{-5}$ | $6.10 \times 10^{-5}$ | $6.10 \times 10^{-5}$ |
| F12 | $6.10 \times 10^{-5}$ | $6.10 \times 10^{-5}$ | $6.10 \times 10^{-5}$ | $6.10 \times 10^{-5}$ | $6.10 \times 10^{-5}$ | $6.10 \times 10^{-5}$ | $8.36 \times 10^{-3}$ |
| F13 | $6.10 \times 10^{-5}$ | $6.10 \times 10^{-5}$ | $6.10 \times 10^{-5}$ | $6.10 \times 10^{-5}$ | $6.10 \times 10^{-5}$ | $6.10 \times 10^{-5}$ | $1.22 \times 10^{-4}$ |
| F14 | $6.10 \times 10^{-5}$ | $6.10 \times 10^{-5}$ | $6.10 \times 10^{-5}$ | $6.10 \times 10^{-5}$ | $8.14 \times 10^{-2}$ | $6.10 \times 10^{-5}$ | $7.93 \times 10^{-3}$ |
| F15 | $3.30 \times 10^{-1}$ | $5.54 \times 10^{-2}$ | $4.54 \times 10^{-1}$ | $5.54 \times 10^{-2}$ | $6.10 \times 10^{-5}$ | $1.16 \times 10^{-3}$ | $1.22 \times 10^{-4}$ |
| F16 | $2.56 \times 10^{-2}$ | $3.36 \times 10^{-3}$ | $6.10 \times 10^{-5}$ | $2.52 \times 10^{-1}$ | $6.10 \times 10^{-5}$ | $6.10 \times 10^{-5}$ | $6.10 \times 10^{-5}$ |
| F17 | $6.10 \times 10^{-4}$ | $6.39 \times 10^{-1}$ | $6.10 \times 10^{-5}$ | $6.10 \times 10^{-5}$ | $6.10 \times 10^{-5}$ | $6.10 \times 10^{-5}$ | $6.10 \times 10^{-5}$ |
| F18 | $1.81 \times 10^{-2}$ | $7.62 \times 10^{-1}$ | $8.36 \times 10^{-3}$ | $8.36 \times 10^{-3}$ | $6.10 \times 10^{-5}$ | $8.36 \times 10^{-3}$ | $6.10 \times 10^{-5}$ |
| F19 | $6.10 \times 10^{-5}$ | $1.03 \times 10^{-2}$ | $2.56 \times 10^{-2}$ | $6.10 \times 10^{-5}$ | $6.10 \times 10^{-5}$ | $4.27 \times 10^{-3}$ | $6.10 \times 10^{-5}$ |
| F20 | $6.39 \times 10^{-1}$ | $1.81 \times 10^{-2}$ | $2.52 \times 10^{-1}$ | $2.52 \times 10^{-1}$ | $1.51 \times 10^{-2}$ | $5.99 \times 10^{-1}$ | $2.08 \times 10^{-1}$ |
| F21 | $3.05 \times 10^{-4}$ | $3.02 \times 10^{-2}$ | $6.10 \times 10^{-5}$ | $1.22 \times 10^{-4}$ | $1.88 \times 10^{-1}$ | $1.53 \times 10^{-3}$ | $8.04 \times 10^{-1}$ |
| F22 | $6.10 \times 10^{-5}$ | $4.27 \times 10^{-3}$ | $6.10 \times 10^{-4}$ | $1.22 \times 10^{-4}$ | $8.04 \times 10^{-1}$ | $3.03 \times 10^{-1}$ | $7.62 \times 10^{-1}$ |
| F23 | $6.10 \times 10^{-4}$ | $1.21 \times 10^{-1}$ | $1.22 \times 10^{-4}$ | $6.10 \times 10^{-5}$ | $3.30 \times 10^{-1}$ | $6.79 \times 10^{-1}$ | $6.79 \times 10^{-1}$ |

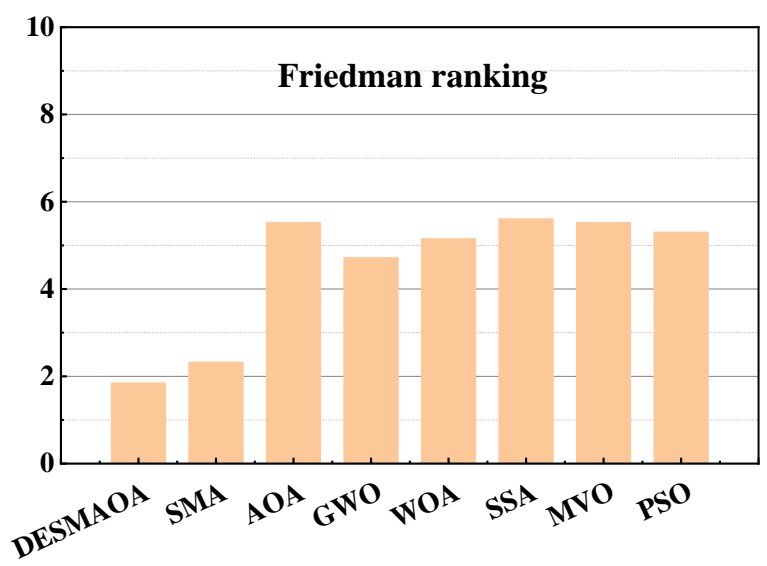

**Figure 2.** Average Friedman ranking values of DESMAOA and other comparative algorithms on 30 dimensions.

### 4.2.2. Qualitative Analysis

Figure 3 shows the qualitative results of proposed algorithm in F4, F5, F6, F8, F12, F13, F15, and F21. From the scatter plot of the search history, we were able to see that search agents were distributed in the whole search space in the early stage. During the iteration progresses, they concentrated in a quick time. The density of distribution for different functions indicates that DESMAOA had balanced performance between exploration and exploitation. Moreover, some sudden changes in the amplitude were observed clearly

in the trajectory of the first search agent, which revealed that DESMAOA had strong exploration capability over the course of iterations when handing these test functions. The drastic fluctuation of average fitness also showed that DESMAOA can jump out of local optima and explore more spaces when dealing with different types of optimization issues. Hence, the local optimal solution can be avoided effectively. Finally, the DESMAOA was able to find better solutions in most of the functions compared with SMA and AOA, which demonstrates the effectiveness of the proposed method.

### 4.2.3. Analysis of Convergence Behavior

It is important to study the convergence behavior of optimization algorithms when they are searching for the optimal solution. In general, fast convergence speed is required in the early exploration, which implies the algorithm has powerful exploration capability. On the other hand, local optima also should be avoided, which can be seen from the convergence curve. Figure 4 shows the convergence curves of DESMAOA and other compared algorithms on 30 dimensions. Some benchmark functions are used for analysis including F4, F5, F6, F8, F12, F13, F15, and F21. From these functions, it can be seen that the initial convergence speed of DESMAOA is the fastest in most cases. In Figure 5, step-like or cliff-like declines in the convergence curves of DESMAOA can be observed. This suggests that the DESMAOA has a prominent exploration capability. From F5, F6, F12, and F13, the precision of solutions for DESMAOA is further improved with the help of SAS during the iteration. In sum, DESMAOA achieved the best solutions in these functions.

### 4.2.4. Scalability Test

The performance fluctuations of optimization algorithms can be revealed according to the scalability test. In this work, the performance of DESMAOA in different dimensions ($D$ = 50, 200, 1000) were also tested. It is easy to understand that the higher dimension will make it harder for the algorithm to find the global optimal solution. Note that only F1–F13 in the 23 benchmark functions were selected for this test. As mentioned previously, F1–F7 are single-mode functions that only have one locally optimal solution. In contrast, F8–F13 are multimode functions that have many locally optimal solutions. Moreover, the experimental parameters were kept the same as previous experiments. Tables 7 and 8 show the results of DESMAOA and other algorithms in different dimensions.

Both results of unimodal and multimode functions indicated that DESMAOA had excellent performance in the conditions of high dimensions. Compared with SMA, AOA, and other well-known algorithms, DESMAOA was the first in all functions except F7. In F7, AOA became better and had more stable results in different dimensions. It is also noted that these comparative algorithms (GWO, WOA, SSA, MVO, and PSO) presented poor optimization capability in some cases, especially in higher dimensions. Furthermore, the Wilcoxon signed-rank test and Friedman ranking test were utilized to analyze the differences between DESMAOA and other algorithms, as listed in Tables 9–12. From Tables 9–11, it can be seen that DESMAOA had significant differences compared with these comparative algorithms. Moreover, in Table 12, the proposed DESMAOA ranked first compared to other algorithms in different dimensions. It is noted that the distance between the first and second was evident. In summary, the proposed DESMAOA had better optimization behavior and stability in dealing with high-dimensional problems.

### 4.3. The IEEE CEC2021 Standard Test Functions

This section describes the IEEE CEC2021 test functions that were employed to further analyze the performance of proposed DESMAOA on solving global optimization problems. The comparative algorithms included the SMA, AOA, GWO, WOA, SSA, MVO, and PSO. To achieve the statistical results, 30 repeated independent tests were conducted for each function. The experimental results are given in Table 13.

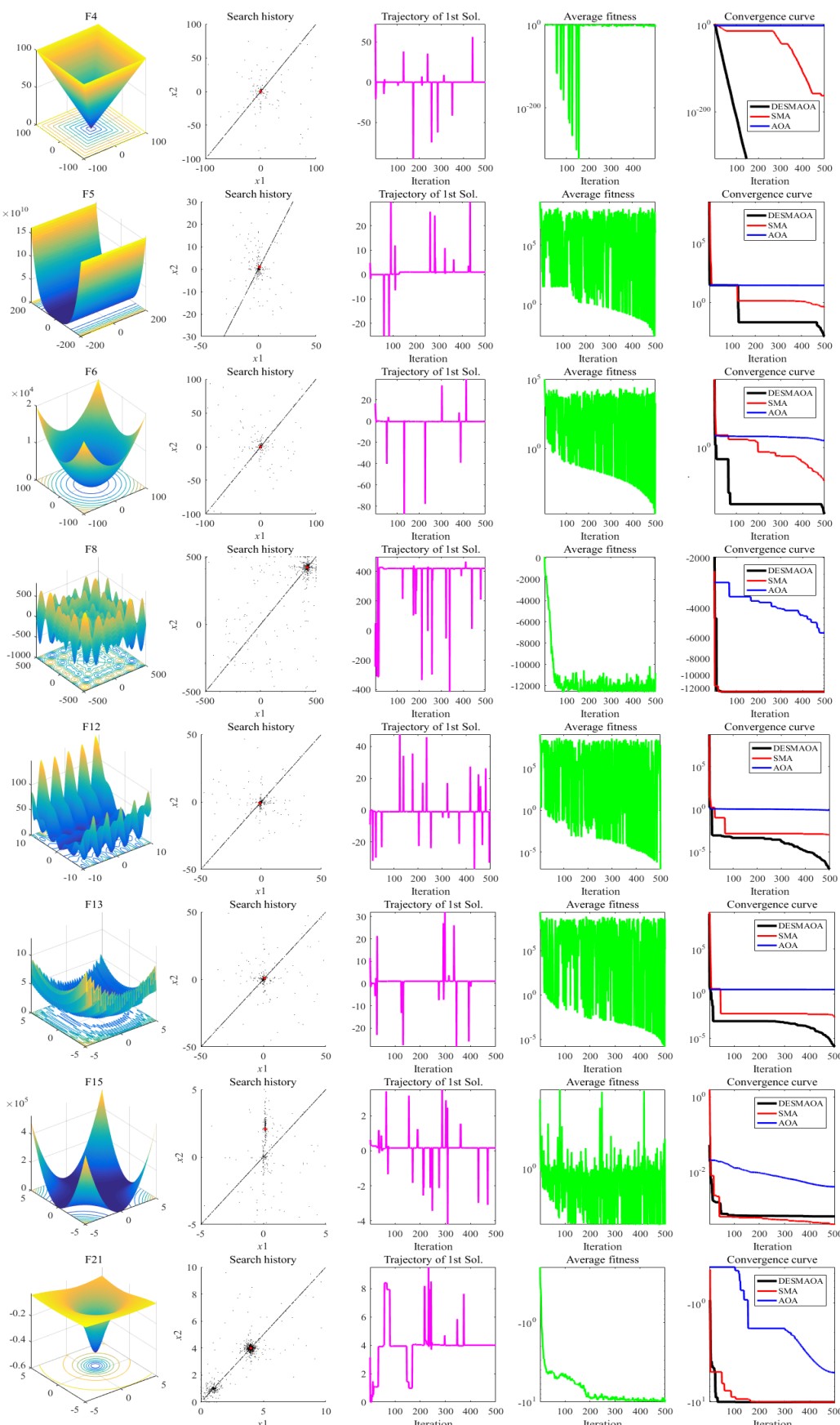

**Figure 3.** Qualitative results for the benchmark functions F4, F5, F6, F8, F12, F13, F15, and F21.

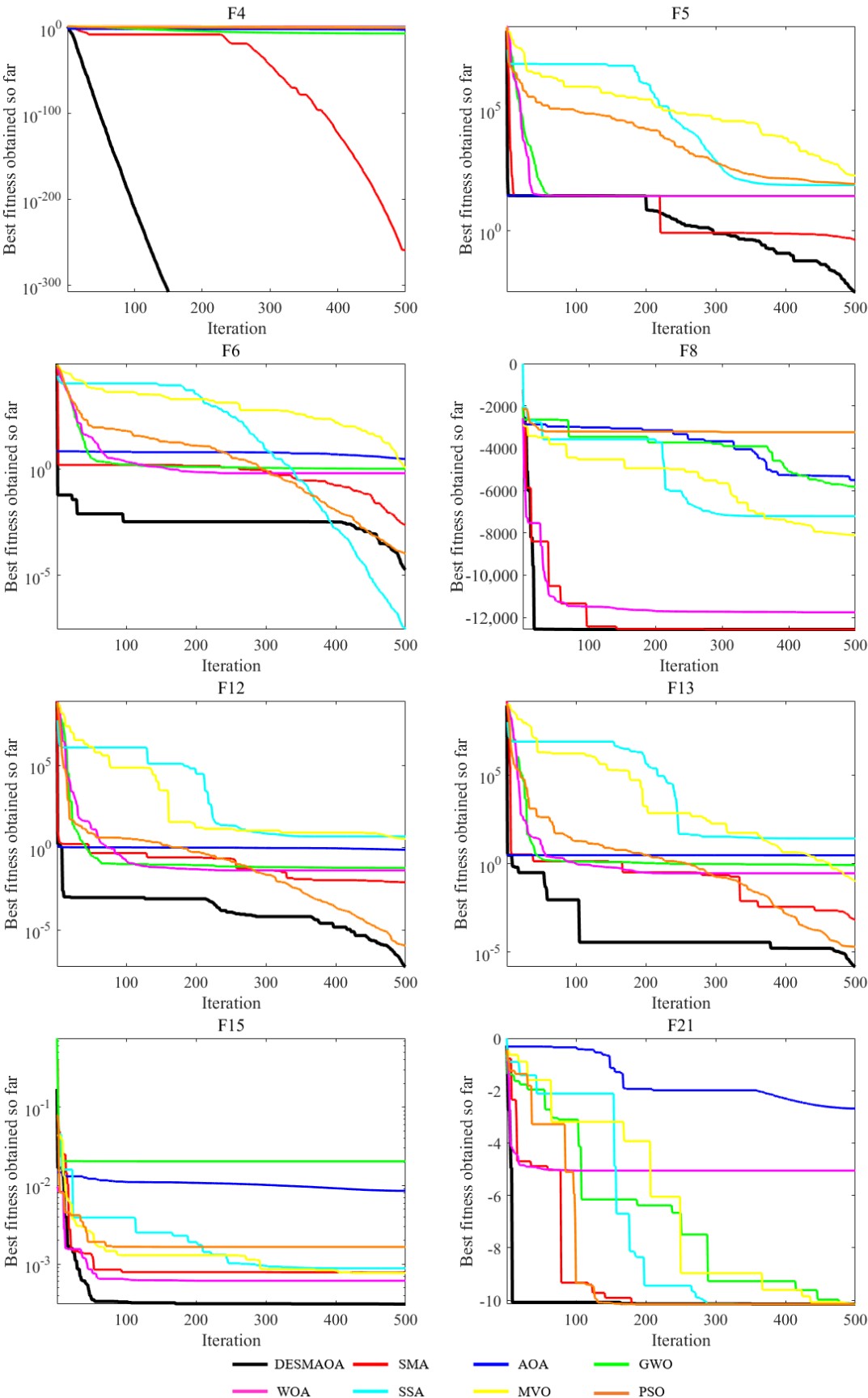

**Figure 4.** The convergence curves of F4, F5, F6, F8, F12, F13, F15, and F21.

**Table 7.** Unimodal benchmark function result statistics of the DESMAOA and competitor algorithms in different dimensions.

| Function | D | Metric | DESMAOA | SMA | AOA | GWO | WOA | SSA | MVO | PSO |
|---|---|---|---|---|---|---|---|---|---|---|
| F1 | 50 | Mean | $0.00 \times 10^{0}$ | $3.94 \times 10^{-310}$ | $4.05 \times 10^{-5}$ | $5.96 \times 10^{-20}$ | $3.97 \times 10^{-71}$ | $6.27 \times 10^{-1}$ | $9.07 \times 10^{0}$ | $2.05 \times 10^{-1}$ |
| | | Std | $0.00 \times 10^{0}$ | $0.00 \times 10^{0}$ | $1.33 \times 10^{-5}$ | $5.86 \times 10^{-20}$ | $2.17 \times 10^{-70}$ | $5.32 \times 10^{-1}$ | $2.48 \times 10^{0}$ | $1.82 \times 10^{-1}$ |
| | 200 | Mean | $0.00 \times 10^{0}$ | $5.15 \times 10^{-244}$ | $4.63 \times 10^{-2}$ | $1.09 \times 10^{-7}$ | $2.33 \times 10^{-71}$ | $1.76 \times 10^{4}$ | $2.84 \times 10^{3}$ | $3.31 \times 10^{2}$ |
| | | Std | $0.00 \times 10^{0}$ | $0.00 \times 10^{0}$ | $1.24 \times 10^{-2}$ | $7.08 \times 10^{-8}$ | $9.24 \times 10^{-71}$ | $1.60 \times 10^{3}$ | $3.15 \times 10^{2}$ | $4.11 \times 10^{1}$ |
| | 1000 | Mean | $0.00 \times 10^{0}$ | $2.20 \times 10^{-246}$ | $1.50 \times 10^{0}$ | $2.53 \times 10^{-1}$ | $3.57 \times 10^{-68}$ | $2.29 \times 10^{5}$ | $7.94 \times 10^{5}$ | $4.11 \times 10^{4}$ |
| | | Std | $0.00 \times 10^{0}$ | $0.00 \times 10^{0}$ | $4.79 \times 10^{-2}$ | $5.65 \times 10^{-2}$ | $1.95 \times 10^{-67}$ | $1.19 \times 10^{4}$ | $2.70 \times 10^{4}$ | $2.32 \times 10^{3}$ |
| F2 | 50 | Mean | $0.00 \times 10^{0}$ | $1.50 \times 10^{-145}$ | $6.70 \times 10^{-3}$ | $2.60 \times 10^{-12}$ | $1.56 \times 10^{-49}$ | $9.29 \times 10^{0}$ | $3.50 \times 10^{3}$ | $2.58 \times 10^{1}$ |
| | | Std | $0.00 \times 10^{0}$ | $8.24 \times 10^{-145}$ | $3.05 \times 10^{-3}$ | $1.52 \times 10^{-12}$ | $7.19 \times 10^{-49}$ | $3.59 \times 10^{0}$ | $1.73 \times 10^{4}$ | $1.89 \times 10^{1}$ |
| | 200 | Mean | $0.00 \times 10^{0}$ | $7.90 \times 10^{-138}$ | $7.23 \times 10^{-2}$ | $3.25 \times 10^{-5}$ | $2.93 \times 10^{-48}$ | $1.55 \times 10^{2}$ | $5.08 \times 10^{77}$ | $4.66 \times 10^{2}$ |
| | | Std | $0.00 \times 10^{0}$ | $3.91 \times 10^{-137}$ | $1.18 \times 10^{-2}$ | $7.69 \times 10^{-6}$ | $1.17 \times 10^{-47}$ | $1.44 \times 10^{1}$ | $2.73 \times 10^{78}$ | $6.40 \times 10^{1}$ |
| | 1000 | Mean | $0.00 \times 10^{0}$ | $5.92 \times 10^{-1}$ | $1.58 \times 10^{0}$ | $6.78 \times 10^{-1}$ | $1.44 \times 10^{-47}$ | $1.19 \times 10^{3}$ | $3.59 \times 10^{278}$ | $1.41 \times 10^{3}$ |
| | | Std | $0.00 \times 10^{0}$ | $2.92 \times 10^{0}$ | $1.08 \times 10^{-1}$ | $5.77 \times 10^{-1}$ | $7.74 \times 10^{-47}$ | $2.48 \times 10^{1}$ | Inf | $6.51 \times 10^{1}$ |
| F3 | 50 | Mean | $0.00 \times 10^{0}$ | $1.03 \times 10^{-293}$ | $1.81 \times 10^{-2}$ | $3.84 \times 10^{-1}$ | $2.01 \times 10^{5}$ | $9.10 \times 10^{3}$ | $6.50 \times 10^{3}$ | $1.48 \times 10^{3}$ |
| | | Std | $0.00 \times 10^{0}$ | $0.00 \times 10^{0}$ | $8.60 \times 10^{-3}$ | $1.01 \times 10^{0}$ | $4.50 \times 10^{4}$ | $4.69 \times 10^{3}$ | $1.95 \times 10^{3}$ | $4.64 \times 10^{2}$ |
| | 200 | Mean | $0.00 \times 10^{0}$ | $3.94 \times 10^{-219}$ | $7.32 \times 10^{-1}$ | $1.98 \times 10^{4}$ | $4.55 \times 10^{6}$ | $2.05 \times 10^{5}$ | $3.16 \times 10^{5}$ | $8.34 \times 10^{4}$ |
| | | Std | $0.00 \times 10^{0}$ | $0.00 \times 10^{0}$ | $1.86 \times 10^{-1}$ | $9.23 \times 10^{3}$ | $1.51 \times 10^{6}$ | $7.25 \times 10^{4}$ | $3.10 \times 10^{4}$ | $2.24 \times 10^{4}$ |
| | 1000 | Mean | $0.00 \times 10^{0}$ | $4.81 \times 10^{-125}$ | $3.35 \times 10^{1}$ | $1.53 \times 10^{6}$ | $1.36 \times 10^{8}$ | $5.19 \times 10^{6}$ | $7.98 \times 10^{6}$ | $2.27 \times 10^{6}$ |
| | | Std | $0.00 \times 10^{0}$ | $2.64 \times 10^{-124}$ | $6.49 \times 10^{0}$ | $3.16 \times 10^{5}$ | $6.32 \times 10^{7}$ | $2.46 \times 10^{6}$ | $8.76 \times 10^{5}$ | $4.56 \times 10^{5}$ |
| F4 | 50 | Mean | $0.00 \times 10^{0}$ | $3.80 \times 10^{-158}$ | $3.56 \times 10^{-2}$ | $7.25 \times 10^{-4}$ | $7.35 \times 10^{1}$ | $1.94 \times 10^{1}$ | $1.67 \times 10^{1}$ | $3.74 \times 10^{0}$ |
| | | Std | $0.00 \times 10^{0}$ | $2.06 \times 10^{-157}$ | $7.14 \times 10^{-3}$ | $1.42 \times 10^{-3}$ | $1.99 \times 10^{1}$ | $3.14 \times 10^{0}$ | $4.24 \times 10^{0}$ | $7.18 \times 10^{-1}$ |
| | 200 | Mean | $0.00 \times 10^{0}$ | $3.11 \times 10^{-114}$ | $9.10 \times 10^{-2}$ | $2.39 \times 10^{1}$ | $8.41 \times 10^{1}$ | $3.52 \times 10^{1}$ | $8.32 \times 10^{1}$ | $1.93 \times 10^{1}$ |
| | | Std | $0.00 \times 10^{0}$ | $1.19 \times 10^{-113}$ | $1.18 \times 10^{-2}$ | $5.51 \times 10^{0}$ | $1.89 \times 10^{1}$ | $3.49 \times 10^{0}$ | $3.80 \times 10^{0}$ | $1.45 \times 10^{0}$ |
| | 1000 | Mean | $0.00 \times 10^{0}$ | $3.86 \times 10^{-101}$ | $1.54 \times 10^{-1}$ | $7.88 \times 10^{1}$ | $7.94 \times 10^{1}$ | $4.43 \times 10^{1}$ | $9.81 \times 10^{1}$ | $3.31 \times 10^{1}$ |
| | | Std | $0.00 \times 10^{0}$ | $1.90 \times 10^{-100}$ | $7.55 \times 10^{-3}$ | $3.25 \times 10^{0}$ | $2.09 \times 10^{1}$ | $3.19 \times 10^{0}$ | $6.42 \times 10^{-1}$ | $1.52 \times 10^{0}$ |
| F5 | 50 | Mean | $4.84 \times 10^{0}$ | $1.89 \times 10^{1}$ | $4.83 \times 10^{1}$ | $4.72 \times 10^{1}$ | $4.83 \times 10^{1}$ | $1.64 \times 10^{3}$ | $7.66 \times 10^{2}$ | $4.21 \times 10^{2}$ |
| | | Std | $1.47 \times 10^{1}$ | $1.93 \times 10^{1}$ | $1.40 \times 10^{-1}$ | $7.35 \times 10^{-1}$ | $4.05 \times 10^{-1}$ | $3.66 \times 10^{3}$ | $7.47 \times 10^{2}$ | $2.18 \times 10^{2}$ |
| | 200 | Mean | $1.29 \times 10^{1}$ | $6.23 \times 10^{1}$ | $1.98 \times 10^{2}$ | $1.98 \times 10^{2}$ | $1.98 \times 10^{2}$ | $3.79 \times 10^{6}$ | $3.91 \times 10^{6}$ | $5.98 \times 10^{5}$ |
| | | Std | $3.68 \times 10^{1}$ | $7.20 \times 10^{1}$ | $7.40 \times 10^{-2}$ | $4.19 \times 10^{-1}$ | $1.65 \times 10^{-1}$ | $9.40 \times 10^{5}$ | $1.21 \times 10^{5}$ | $1.04 \times 10^{5}$ |
| | 1000 | Mean | $1.11 \times 10^{2}$ | $4.01 \times 10^{2}$ | $1.00 \times 10^{3}$ | $1.05 \times 10^{3}$ | $9.94 \times 10^{2}$ | $1.21 \times 10^{8}$ | $2.33 \times 10^{9}$ | $2.95 \times 10^{8}$ |
| | | Std | $2.50 \times 10^{2}$ | $4.15 \times 10^{2}$ | $2.71 \times 10^{-1}$ | $2.55 \times 10^{1}$ | $1.03 \times 10^{0}$ | $1.12 \times 10^{7}$ | $1.85 \times 10^{8}$ | $4.28 \times 10^{7}$ |
| F6 | 50 | Mean | $1.66 \times 10^{-4}$ | $8.78 \times 10^{-2}$ | $7.29 \times 10^{0}$ | $2.57 \times 10^{0}$ | $1.20 \times 10^{0}$ | $8.30 \times 10^{-1}$ | $9.61 \times 10^{0}$ | $1.97 \times 10^{-1}$ |
| | | Std | $5.60 \times 10^{-5}$ | $6.54 \times 10^{-2}$ | $4.04 \times 10^{-1}$ | $4.02 \times 10^{-1}$ | $5.39 \times 10^{-1}$ | $7.25 \times 10^{-1}$ | $1.88 \times 10^{0}$ | $1.74 \times 10^{-1}$ |
| | 200 | Mean | $3.73 \times 10^{-2}$ | $8.26 \times 10^{0}$ | $3.59 \times 10^{1}$ | $2.91 \times 10^{1}$ | $1.11 \times 10^{1}$ | $1.76 \times 10^{4}$ | $2.99 \times 10^{3}$ | $3.21 \times 10^{2}$ |
| | | Std | $3.21 \times 10^{-2}$ | $8.04 \times 10^{0}$ | $1.19 \times 10^{0}$ | $1.28 \times 10^{0}$ | $2.90 \times 10^{0}$ | $2.45 \times 10^{3}$ | $4.10 \times 10^{2}$ | $4.66 \times 10^{1}$ |
| | 1000 | Mean | $3.34 \times 10^{0}$ | $6.80 \times 10^{1}$ | $2.42 \times 10^{2}$ | $2.02 \times 10^{2}$ | $6.68 \times 10^{1}$ | $2.37 \times 10^{5}$ | $8.04 \times 10^{5}$ | $4.02 \times 10^{4}$ |
| | | Std | $4.80 \times 10^{0}$ | $8.81 \times 10^{1}$ | $1.23 \times 10^{0}$ | $2.57 \times 10^{0}$ | $1.52 \times 10^{1}$ | $1.11 \times 10^{4}$ | $2.79 \times 10^{4}$ | $2.17 \times 10^{3}$ |
| F7 | 50 | Mean | $1.36 \times 10^{-4}$ | $1.96 \times 10^{-4}$ | $6.63 \times 10^{-5}$ | $3.54 \times 10^{-3}$ | $3.97 \times 10^{-3}$ | $4.86 \times 10^{-1}$ | $1.07 \times 10^{-1}$ | $3.97 \times 10^{1}$ |
| | | Std | $1.18 \times 10^{-4}$ | $1.69 \times 10^{-4}$ | $5.90 \times 10^{-5}$ | $1.90 \times 10^{-3}$ | $4.79 \times 10^{-3}$ | $1.53 \times 10^{-1}$ | $2.32 \times 10^{-2}$ | $2.76 \times 10^{1}$ |
| | 200 | Mean | $1.29 \times 10^{-4}$ | $4.32 \times 10^{-4}$ | $5.35 \times 10^{-5}$ | $1.63 \times 10^{-2}$ | $4.14 \times 10^{-3}$ | $1.72 \times 10^{1}$ | $5.40 \times 10^{0}$ | $2.95 \times 10^{3}$ |
| | | Std | $1.52 \times 10^{-4}$ | $3.04 \times 10^{-4}$ | $5.09 \times 10^{-5}$ | $5.34 \times 10^{-3}$ | $4.22 \times 10^{-3}$ | $4.14 \times 10^{0}$ | $7.17 \times 10^{-1}$ | $4.68 \times 10^{2}$ |
| | 1000 | Mean | $1.17 \times 10^{-4}$ | $6.93 \times 10^{-4}$ | $8.25 \times 10^{-5}$ | $1.55 \times 10^{-1}$ | $3.29 \times 10^{-3}$ | $1.74 \times 10^{3}$ | $2.88 \times 10^{4}$ | $2.39 \times 10^{5}$ |
| | | Std | $1.28 \times 10^{-4}$ | $4.85 \times 10^{-4}$ | $6.96 \times 10^{-5}$ | $3.32 \times 10^{-2}$ | $3.73 \times 10^{-3}$ | $1.75 \times 10^{2}$ | $2.72 \times 10^{3}$ | $7.66 \times 10^{3}$ |

**Table 8.** Unimodal benchmark function result statistics of the DESMAOA and competitor algorithms in different dimensions.

| Function | D | Metric | DESMAOA | SMA | AOA | GWO | WOA | SSA | MVO | PSO |
|---|---|---|---|---|---|---|---|---|---|---|
| F8 | 50 | Mean | $-2.0949 \times 10^4$ | $-2.0947 \times 10^4$ | $-8.3989 \times 10^3$ | $-9.1468 \times 10^3$ | $-1.8030 \times 10^4$ | $-1.2107 \times 10^4$ | $-1.2350 \times 10^4$ | $-7.6172 \times 10^3$ |
| | | Std | $1.54 \times 10^{-4}$ | $2.27 \times 10^0$ | $5.06 \times 10^2$ | $1.59 \times 10^3$ | $2.78 \times 10^3$ | $1.00 \times 10^3$ | $1.11 \times 10^3$ | $2.18 \times 10^3$ |
| | 200 | Mean | $-8.3796 \times 10^4$ | $-8.3757 \times 10^4$ | $-2.1657 \times 10^4$ | $-2.7533 \times 10^4$ | $-7.1281 \times 10^4$ | $-3.4381 \times 10^4$ | $-4.0399 \times 10^4$ | $-1.5591 \times 10^4$ |
| | | Std | $6.62 \times 10^{-1}$ | $6.42 \times 10^1$ | $1.27 \times 10^3$ | $5.65 \times 10^3$ | $1.28 \times 10^4$ | $2.25 \times 10^3$ | $2.30 \times 10^3$ | $6.41 \times 10^3$ |
| | 1000 | Mean | $-4.1892 \times 10^5$ | $-4.1862 \times 10^5$ | $-5.4566 \times 10^4$ | $-8.4602 \times 10^4$ | $-3.5941 \times 10^5$ | $-8.8084 \times 10^4$ | $-1.1041 \times 10^5$ | $-3.3236 \times 10^4$ |
| | | Std | $1.25 \times 10^2$ | $5.75 \times 10^2$ | $2.29 \times 10^3$ | $2.28 \times 10^4$ | $5.92 \times 10^4$ | $7.25 \times 10^3$ | $3.94 \times 10^3$ | $1.51 \times 10^4$ |
| F9 | 50 | Mean | $0.00 \times 10^0$ | $0.00 \times 10^0$ | $1.61 \times 10^{-5}$ | $5.24 \times 10^0$ | $1.89 \times 10^{-15}$ | $8.82 \times 10^1$ | $2.54 \times 10^2$ | $2.84 \times 10^2$ |
| | | Std | $0.00 \times 10^0$ | $0.00 \times 10^0$ | $4.42 \times 10^{-6}$ | $7.71 \times 10^0$ | $1.04 \times 10^{-14}$ | $2.32 \times 10^1$ | $5.65 \times 10^1$ | $5.01 \times 10^1$ |
| | 200 | Mean | $0.00 \times 10^0$ | $0.00 \times 10^0$ | $1.34 \times 10^{-3}$ | $2.41 \times 10^1$ | $7.58 \times 10^{-15}$ | $8.27 \times 10^2$ | $1.90 \times 10^3$ | $2.02 \times 10^3$ |
| | | Std | $0.00 \times 10^0$ | $0.00 \times 10^0$ | $1.82 \times 10^{-4}$ | $9.14 \times 10^0$ | $4.15 \times 10^{-14}$ | $8.74 \times 10^1$ | $1.30 \times 10^2$ | $1.25 \times 10^2$ |
| | 1000 | Mean | $0.00 \times 10^0$ | $0.00 \times 10^0$ | $3.79 \times 10^{-2}$ | $2.06 \times 10^2$ | $0.00 \times 10^0$ | $7.63 \times 10^3$ | $1.46 \times 10^4$ | $1.41 \times 10^4$ |
| | | Std | $0.00 \times 10^0$ | $0.00 \times 10^0$ | $1.94 \times 10^{-3}$ | $5.67 \times 10^1$ | $0.00 \times 10^0$ | $2.12 \times 10^2$ | $2.44 \times 10^2$ | $2.98 \times 10^2$ |
| F10 | 50 | Mean | $8.8818 \times 10^{-16}$ | $8.8818 \times 10^{-16}$ | $1.14 \times 10^{-3}$ | $4.3720 \times 10^{-11}$ | $4.3225 \times 10^{-15}$ | $4.83 \times 10^0$ | $3.56 \times 10^0$ | $1.69 \times 10^0$ |
| | | Std | $0.00 \times 10^0$ | $0.00 \times 10^0$ | $1.93 \times 10^{-4}$ | $2.44 \times 10^{-11}$ | $2.38 \times 10^{-15}$ | $1.23 \times 10^0$ | $3.13 \times 10^0$ | $5.70 \times 10^{-1}$ |
| | 200 | Mean | $8.8818 \times 10^{-16}$ | $8.8818 \times 10^{-16}$ | $1.06 \times 10^{-2}$ | $2.18 \times 10^{-5}$ | $4.09 \times 10^{-15}$ | $1.30 \times 10^1$ | $2.04 \times 10^1$ | $6.61 \times 10^0$ |
| | | Std | $0.00 \times 10^0$ | $0.00 \times 10^0$ | $1.01 \times 10^{-3}$ | $6.01 \times 10^{-6}$ | $2.70 \times 10^{-15}$ | $4.61 \times 10^{-1}$ | $2.15 \times 10^{-1}$ | $3.38 \times 10^{-1}$ |
| | 1000 | Mean | $8.8818 \times 10^{-16}$ | $8.8818 \times 10^{-16}$ | $3.32 \times 10^{-2}$ | $1.89 \times 10^{-2}$ | $4.91 \times 10^{-15}$ | $1.45 \times 10^1$ | $2.10 \times 10^1$ | $1.60 \times 10^1$ |
| | | Std | $0.00 \times 10^0$ | $0.00 \times 10^0$ | $7.69 \times 10^{-4}$ | $3.23 \times 10^{-3}$ | $2.42 \times 10^{-15}$ | $1.94 \times 10^{-1}$ | $3.27 \times 10^{-2}$ | $2.33 \times 10^{-1}$ |
| F11 | 50 | Mean | $0.00 \times 10^0$ | $0.00 \times 10^0$ | $7.70 \times 10^{-3}$ | $2.94 \times 10^{-3}$ | $1.34 \times 10^{-2}$ | $5.55 \times 10^{-1}$ | $1.09 \times 10^0$ | $1.62 \times 10^{-2}$ |
| | | Std | $0.00 \times 10^0$ | $0.00 \times 10^0$ | $1.91 \times 10^{-2}$ | $6.06 \times 10^{-3}$ | $5.11 \times 10^{-2}$ | $2.74 \times 10^{-1}$ | $2.30 \times 10^{-2}$ | $1.19 \times 10^{-2}$ |
| | 200 | Mean | $0.00 \times 10^0$ | $0.00 \times 10^0$ | $7.85 \times 10^0$ | $6.27 \times 10^{-3}$ | $0.00 \times 10^0$ | $1.46 \times 10^2$ | $2.73 \times 10^1$ | $2.28 \times 10^0$ |
| | | Std | $0.00 \times 10^0$ | $0.00 \times 10^0$ | $1.19 \times 10^1$ | $1.48 \times 10^{-2}$ | $0.00 \times 10^0$ | $1.88 \times 10^1$ | $3.08 \times 10^0$ | $2.70 \times 10^0$ |
| | 1000 | Mean | $0.00 \times 10^0$ | $0.00 \times 10^0$ | $1.33 \times 10^4$ | $2.37 \times 10^{-2}$ | $0.00 \times 10^0$ | $2.12 \times 10^3$ | $7.25 \times 10^3$ | $2.74 \times 10^2$ |
| | | Std | $0.00 \times 10^0$ | $0.00 \times 10^0$ | $2.64 \times 10^3$ | $3.67 \times 10^{-2}$ | $0.00 \times 10^0$ | $8.35 \times 10^1$ | $3.01 \times 10^2$ | $1.86 \times 10^1$ |
| F12 | 50 | Mean | $6.67 \times 10^{-7}$ | $6.02 \times 10^{-3}$ | $9.06 \times 10^{-1}$ | $1.22 \times 10^{-1}$ | $3.46 \times 10^{-2}$ | $1.26 \times 10^1$ | $5.51 \times 10^0$ | $8.03 \times 10^{-2}$ |
| | | Std | $6.28 \times 10^{-7}$ | $1.22 \times 10^{-2}$ | $2.39 \times 10^{-2}$ | $7.35 \times 10^{-2}$ | $1.86 \times 10^{-2}$ | $4.57 \times 10^0$ | $1.37 \times 10^0$ | $1.53 \times 10^{-1}$ |
| | 200 | Mean | $2.01 \times 10^{-5}$ | $5.76 \times 10^{-3}$ | $8.41 \times 10^{-1}$ | $5.42 \times 10^{-1}$ | $7.03 \times 10^{-2}$ | $7.55 \times 10^3$ | $2.30 \times 10^3$ | $4.84 \times 10^1$ |
| | | Std | $1.57 \times 10^{-5}$ | $8.11 \times 10^{-3}$ | $5.60 \times 10^{-2}$ | $6.68 \times 10^{-2}$ | $3.52 \times 10^{-2}$ | $1.01 \times 10^4$ | $2.88 \times 10^3$ | $3.71 \times 10^1$ |
| | 1000 | Mean | $1.53 \times 10^{-4}$ | $9.67 \times 10^{-3}$ | $1.04 \times 10^0$ | $1.26 \times 10^0$ | $1.05 \times 10^{-1}$ | $1.16 \times 10^7$ | $4.19 \times 10^9$ | $9.21 \times 10^6$ |
| | | Std | $2.46 \times 10^{-4}$ | $1.70 \times 10^{-2}$ | $1.12 \times 10^{-2}$ | $2.99 \times 10^{-1}$ | $5.30 \times 10^{-2}$ | $4.67 \times 10^6$ | $4.67 \times 10^8$ | $2.30 \times 10^6$ |
| F13 | 50 | Mean | $1.08 \times 10^{-3}$ | $2.52 \times 10^{-2}$ | $4.94 \times 10^0$ | $2.03 \times 10^0$ | $1.14 \times 10^0$ | $8.07 \times 10^1$ | $7.29 \times 10^0$ | $1.84 \times 10^{-1}$ |
| | | Std | $4.27 \times 10^{-3}$ | $3.02 \times 10^{-2}$ | $6.56 \times 10^{-4}$ | $2.80 \times 10^{-1}$ | $4.84 \times 10^{-1}$ | $1.61 \times 10^1$ | $1.15 \times 10^1$ | $1.16 \times 10^{-1}$ |
| | 200 | Mean | $1.85 \times 10^{-3}$ | $4.73 \times 10^{-1}$ | $1.97 \times 10^1$ | $1.67 \times 10^1$ | $6.18 \times 10^0$ | $1.61 \times 10^6$ | $1.11 \times 10^5$ | $5.27 \times 10^3$ |
| | | Std | $1.29 \times 10^{-3}$ | $7.17 \times 10^{-1}$ | $9.97 \times 10^{-2}$ | $4.32 \times 10^{-1}$ | $1.75 \times 10^0$ | $7.60 \times 10^5$ | $1.03 \times 10^5$ | $2.58 \times 10^3$ |
| | 1000 | Mean | $6.06 \times 10^{-2}$ | $3.82 \times 10^0$ | $1.00 \times 10^2$ | $1.21 \times 10^2$ | $4.02 \times 10^1$ | $1.47 \times 10^8$ | $9.13 \times 10^9$ | $8.26 \times 10^7$ |
| | | Std | $8.56 \times 10^{-2}$ | $3.59 \times 10^0$ | $3.49 \times 10^{-1}$ | $7.98 \times 10^0$ | $1.12 \times 10^1$ | $2.91 \times 10^7$ | $8.64 \times 10^8$ | $1.28 \times 10^7$ |

**Table 9.** *p*-values of the Wilcoxon signed-rank test between DESMAOA and other competitor algorithms on 50 dimensions.

| Function | DESMAOA vs. SMA | DESMAOA vs. AOA | DESMAOA vs. GWO | DESMAOA vs. WOA | DESMAOA vs. SSA | DESMAOA vs. MVO | DESMAOA vs. PSO |
|---|---|---|---|---|---|---|---|
| F1 | $5.00 \times 10^{-1}$ | $6.10 \times 10^{-5}$ | $6.10 \times 10^{-5}$ | $6.10 \times 10^{-5}$ | $6.10 \times 10^{-5}$ | $6.10 \times 10^{-5}$ | $6.10 \times 10^{-5}$ |
| F2 | $6.10 \times 10^{-5}$ | $6.10 \times 10^{-5}$ | $6.10 \times 10^{-5}$ | $6.10 \times 10^{-5}$ | $6.10 \times 10^{-5}$ | $6.10 \times 10^{-5}$ | $6.10 \times 10^{-5}$ |
| F3 | $5.00 \times 10^{-1}$ | $6.10 \times 10^{-5}$ | $6.10 \times 10^{-5}$ | $6.10 \times 10^{-5}$ | $6.10 \times 10^{-5}$ | $6.10 \times 10^{-5}$ | $6.10 \times 10^{-5}$ |
| F4 | $6.10 \times 10^{-5}$ | $6.10 \times 10^{-5}$ | $6.10 \times 10^{-5}$ | $6.10 \times 10^{-5}$ | $6.10 \times 10^{-5}$ | $6.10 \times 10^{-5}$ | $6.10 \times 10^{-5}$ |
| F5 | $5.37 \times 10^{-3}$ | $6.10 \times 10^{-5}$ | $1.22 \times 10^{-4}$ | $6.10 \times 10^{-5}$ | $6.10 \times 10^{-5}$ | $6.10 \times 10^{-5}$ | $6.10 \times 10^{-5}$ |
| F6 | $6.10 \times 10^{-5}$ | $6.10 \times 10^{-5}$ | $6.10 \times 10^{-5}$ | $6.10 \times 10^{-5}$ | $6.10 \times 10^{-5}$ | $6.10 \times 10^{-5}$ | $6.10 \times 10^{-5}$ |
| F7 | $8.33 \times 10^{-2}$ | $7.30 \times 10^{-2}$ | $6.10 \times 10^{-5}$ | $6.10 \times 10^{-5}$ | $6.10 \times 10^{-5}$ | $6.10 \times 10^{-5}$ | $6.10 \times 10^{-5}$ |
| F8 | $6.10 \times 10^{-5}$ | $6.10 \times 10^{-5}$ | $6.10 \times 10^{-5}$ | $6.10 \times 10^{-5}$ | $6.10 \times 10^{-5}$ | $6.10 \times 10^{-5}$ | $6.10 \times 10^{-5}$ |
| F9 | $1.00 \times 10^{0}$ | $6.10 \times 10^{-5}$ | $6.10 \times 10^{-5}$ | $1.00 \times 10^{0}$ | $6.10 \times 10^{-5}$ | $6.10 \times 10^{-5}$ | $6.10 \times 10^{-5}$ |
| F10 | $1.00 \times 10^{0}$ | $6.10 \times 10^{-5}$ | $6.10 \times 10^{-5}$ | $2.44 \times 10^{-4}$ | $6.10 \times 10^{-5}$ | $6.10 \times 10^{-5}$ | $6.10 \times 10^{-5}$ |
| F11 | $1.00 \times 10^{0}$ | $6.10 \times 10^{-5}$ | $2.50 \times 10^{-1}$ | $1.00 \times 10^{0}$ | $6.10 \times 10^{-5}$ | $6.10 \times 10^{-5}$ | $6.10 \times 10^{-5}$ |
| F12 | $6.10 \times 10^{-5}$ | $6.10 \times 10^{-5}$ | $6.10 \times 10^{-5}$ | $6.10 \times 10^{-5}$ | $6.10 \times 10^{-5}$ | $6.10 \times 10^{-5}$ | $6.10 \times 10^{-5}$ |
| F13 | $2.01 \times 10^{-3}$ | $6.10 \times 10^{-5}$ | $6.10 \times 10^{-5}$ | $6.10 \times 10^{-5}$ | $6.10 \times 10^{-5}$ | $6.10 \times 10^{-5}$ | $6.10 \times 10^{-5}$ |

**Table 10.** *p*-Values of the Wilcoxon signed-rank test between DESMAOA and other competitor algorithms on 200 dimensions.

| Function | DESMAOA vs. SMA | DESMAOA vs. AOA | DESMAOA vs. GWO | DESMAOA vs. WOA | DESMAOA vs. SSA | DESMAOA vs. MVO | DESMAOA vs. PSO |
|---|---|---|---|---|---|---|---|
| F1 | $2.50 \times 10^{-1}$ | $6.10 \times 10^{-5}$ | $6.10 \times 10^{-5}$ | $6.10 \times 10^{-5}$ | $6.10 \times 10^{-5}$ | $6.10 \times 10^{-5}$ | $6.10 \times 10^{-5}$ |
| F2 | $6.10 \times 10^{-5}$ | $6.10 \times 10^{-5}$ | $6.10 \times 10^{-5}$ | $6.10 \times 10^{-5}$ | $6.10 \times 10^{-5}$ | $6.10 \times 10^{-5}$ | $6.10 \times 10^{-5}$ |
| F3 | $6.10 \times 10^{-5}$ | $6.10 \times 10^{-5}$ | $9.77 \times 10^{-4}$ | $6.10 \times 10^{-5}$ | $6.10 \times 10^{-5}$ | $6.10 \times 10^{-5}$ | $6.10 \times 10^{-5}$ |
| F4 | $6.10 \times 10^{-5}$ | $6.10 \times 10^{-5}$ | $6.10 \times 10^{-5}$ | $6.10 \times 10^{-5}$ | $6.10 \times 10^{-5}$ | $6.10 \times 10^{-5}$ | $6.10 \times 10^{-5}$ |
| F5 | $6.10 \times 10^{-5}$ | $6.10 \times 10^{-5}$ | $6.10 \times 10^{-5}$ | $6.10 \times 10^{-5}$ | $8.54 \times 10^{-4}$ | $6.10 \times 10^{-5}$ | $6.10 \times 10^{-5}$ |
| F6 | $6.10 \times 10^{-5}$ | $6.10 \times 10^{-5}$ | $6.10 \times 10^{-5}$ | $6.10 \times 10^{-5}$ | $6.10 \times 10^{-5}$ | $1.22 \times 10^{-4}$ | $6.10 \times 10^{-5}$ |
| F7 | $6.10 \times 10^{-5}$ | $6.10 \times 10^{-5}$ | $6.10 \times 10^{-5}$ | $6.10 \times 10^{-5}$ | $6.10 \times 10^{-5}$ | $6.10 \times 10^{-5}$ | $6.10 \times 10^{-5}$ |
| F8 | $2.56 \times 10^{-2}$ | $6.10 \times 10^{-5}$ | $1.22 \times 10^{-4}$ | $6.10 \times 10^{-5}$ | $6.10 \times 10^{-5}$ | $6.10 \times 10^{-5}$ | $6.10 \times 10^{-5}$ |
| F9 | $1.00 \times 10^{0}$ | $6.10 \times 10^{-5}$ | $6.10 \times 10^{-5}$ | $6.10 \times 10^{-5}$ | $6.10 \times 10^{-5}$ | $6.10 \times 10^{-5}$ | $6.10 \times 10^{-5}$ |
| F10 | $1.00 \times 10^{0}$ | $6.10 \times 10^{-5}$ | $6.10 \times 10^{-5}$ | $6.10 \times 10^{-5}$ | $6.10 \times 10^{-5}$ | $6.10 \times 10^{-5}$ | $6.10 \times 10^{-5}$ |
| F11 | $1.00 \times 10^{0}$ | $6.10 \times 10^{-5}$ | $6.10 \times 10^{-5}$ | $1.00 \times 10^{0}$ | $6.10 \times 10^{-5}$ | $4.88 \times 10^{-4}$ | $6.10 \times 10^{-5}$ |
| F12 | $6.10 \times 10^{-5}$ | $6.10 \times 10^{-5}$ | $6.10 \times 10^{-5}$ | $6.10 \times 10^{-5}$ | $6.10 \times 10^{-5}$ | $6.10 \times 10^{-5}$ | $1.22 \times 10^{-4}$ |
| F13 | $6.10 \times 10^{-5}$ | $6.10 \times 10^{-5}$ | $6.10 \times 10^{-5}$ | $6.10 \times 10^{-5}$ | $6.10 \times 10^{-5}$ | $6.10 \times 10^{-5}$ | $6.10 \times 10^{-5}$ |

**Table 11.** *p*-Values of the Wilcoxon signed-rank test between DESMAOA and other competitor algorithms on 1000 dimensions.

| Function | DESMAOA vs. SMA | DESMAOA vs. AOA | DESMAOA vs. GWO | DESMAOA vs. WOA | DESMAOA vs. SSA | DESMAOA vs. MVO | DESMAOA vs. PSO |
|---|---|---|---|---|---|---|---|
| F1 | $3.91 \times 10^{-3}$ | $6.10 \times 10^{-5}$ | $6.10 \times 10^{-5}$ | $6.10 \times 10^{-5}$ | $6.10 \times 10^{-5}$ | $6.10 \times 10^{-5}$ | $6.10 \times 10^{-5}$ |
| F2 | $6.10 \times 10^{-5}$ | $6.10 \times 10^{-5}$ | $6.10 \times 10^{-5}$ | $6.10 \times 10^{-5}$ | $6.10 \times 10^{-5}$ | $6.10 \times 10^{-5}$ | $6.10 \times 10^{-5}$ |
| F3 | $6.10 \times 10^{-5}$ | $6.10 \times 10^{-5}$ | $1.22 \times 10^{-4}$ | $6.10 \times 10^{-5}$ | $6.10 \times 10^{-5}$ | $6.10 \times 10^{-5}$ | $6.10 \times 10^{-5}$ |
| F4 | $6.10 \times 10^{-5}$ | $6.10 \times 10^{-5}$ | $6.10 \times 10^{-5}$ | $6.10 \times 10^{-5}$ | $6.10 \times 10^{-5}$ | $6.10 \times 10^{-5}$ | $6.10 \times 10^{-5}$ |
| F5 | $6.10 \times 10^{-5}$ | $6.10 \times 10^{-5}$ | $6.10 \times 10^{-5}$ | $6.10 \times 10^{-5}$ | $6.10 \times 10^{-4}$ | $6.10 \times 10^{-5}$ | $6.10 \times 10^{-5}$ |
| F6 | $6.10 \times 10^{-5}$ | $6.10 \times 10^{-5}$ | $6.10 \times 10^{-5}$ | $6.10 \times 10^{-5}$ | $6.10 \times 10^{-5}$ | $1.22 \times 10^{-4}$ | $6.10 \times 10^{-5}$ |
| F7 | $6.10 \times 10^{-5}$ | $6.10 \times 10^{-5}$ | $6.10 \times 10^{-5}$ | $6.10 \times 10^{-5}$ | $6.10 \times 10^{-5}$ | $6.10 \times 10^{-5}$ | $8.36 \times 10^{-3}$ |
| F8 | $4.21 \times 10^{-1}$ | $6.10 \times 10^{-5}$ | $1.22 \times 10^{-4}$ | $6.10 \times 10^{-5}$ | $6.10 \times 10^{-5}$ | $6.10 \times 10^{-5}$ | $6.10 \times 10^{-5}$ |
| F9 | $1.00 \times 10^{0}$ | $6.10 \times 10^{-5}$ | $6.10 \times 10^{-5}$ | $1.00 \times 10^{0}$ | $6.10 \times 10^{-5}$ | $6.10 \times 10^{-5}$ | $6.10 \times 10^{-5}$ |
| F10 | $1.00 \times 10^{0}$ | $6.10 \times 10^{-5}$ | $6.10 \times 10^{-5}$ | $6.10 \times 10^{-5}$ | $6.10 \times 10^{-5}$ | $6.10 \times 10^{-5}$ | $6.10 \times 10^{-5}$ |
| F11 | $1.00 \times 10^{0}$ | $6.10 \times 10^{-5}$ | $6.10 \times 10^{-5}$ | $1.00 \times 10^{0}$ | $6.10 \times 10^{-5}$ | $9.77 \times 10^{-4}$ | $6.10 \times 10^{-5}$ |
| F12 | $6.10 \times 10^{-5}$ | $6.10 \times 10^{-5}$ | $6.10 \times 10^{-5}$ | $6.10 \times 10^{-5}$ | $6.10 \times 10^{-5}$ | $6.10 \times 10^{-5}$ | $1.16 \times 10^{-3}$ |
| F13 | $6.10 \times 10^{-5}$ | $6.10 \times 10^{-5}$ | $6.10 \times 10^{-5}$ | $6.10 \times 10^{-5}$ | $6.10 \times 10^{-5}$ | $6.10 \times 10^{-5}$ | $6.10 \times 10^{-5}$ |

**Table 12.** Experimental results of Friedman test on 50, 200, and 1000 dimensions.

| Algorithm | *D* = 50 | | *D* = 200 | | *D* = 1000 | |
|---|---|---|---|---|---|---|
| | **Mean** | **Rank** | **Mean** | | **Mean** | **Rank** |
| DESMAOA | 1.1923 | 1 | 1.2308 | 1 | 1.2692 | 1 |
| SMA | 1.9615 | 2 | 2.0000 | 2 | 2.1923 | 2 |
| AOA | 4.7692 | 5 | 4.5385 | 5 | 4.4615 | 4 |
| GWO | 4.3077 | 3 | 4.3846 | 4 | 4.6923 | 5 |
| WOA | 4.3846 | 4 | 3.7692 | 3 | 3.4615 | 3 |
| SSA | 6.7692 | 7 | 7.0000 | 8 | 6.1538 | 7 |
| MVO | 6.8462 | 8 | 6.7692 | 7 | 7.4615 | 8 |
| PSO | 5.7692 | 6 | 6.3077 | 6 | 6.3077 | 6 |

**Table 13.** The result statistics of CEC2021 test functions for the DESMAOA and competitor algorithms.

| Function | Metric | DESMAOA | SMA | AOA | GWO | WOA | SSA | MVO | PSO |
|---|---|---|---|---|---|---|---|---|---|
| CEC_01 | Mean | $3.4126 \times 10^3$ | $8.4790 \times 10^3$ | $1.4400 \times 10^{10}$ | $8.5800 \times 10^7$ | $8.8200 \times 10^7$ | $3.0497 \times 10^3$ | $2.0700 \times 10^4$ | $2.6509 \times 10^3$ |
| | Std | $3.2484 \times 10^3$ | $4.3942 \times 10^3$ | $5.4800 \times 10^9$ | $2.0600 \times 10^8$ | $1.0800 \times 10^8$ | $2.6586 \times 10^3$ | $1.2300 \times 10^4$ | $2.8147 \times 10^3$ |
| CEC_02 | Mean | $1.6460 \times 10^3$ | $1.7230 \times 10^3$ | $2.3794 \times 10^3$ | $1.7650 \times 10^3$ | $2.3408 \times 10^3$ | $1.9202 \times 10^3$ | $1.7883 \times 10^3$ | $2.0353 \times 10^3$ |
| | Std | $1.8064 \times 10^2$ | $2.0356 \times 10^2$ | $2.5500 \times 10^2$ | $4.0002 \times 10^2$ | $2.8907 \times 10^2$ | $3.0299 \times 10^2$ | $2.8914 \times 10^2$ | $3.3248 \times 10^2$ |
| CEC_03 | Mean | $7.4197 \times 10^2$ | $7.3268 \times 10^2$ | $8.0255 \times 10^2$ | $7.3186 \times 10^2$ | $7.9430 \times 10^2$ | $7.4133 \times 10^2$ | $7.3257 \times 10^2$ | $7.3096 \times 10^2$ |
| | Std | $1.3713 \times 10^1$ | $9.9242 \times 10^0$ | $8.5999 \times 10^0$ | $1.0358 \times 10^1$ | $2.9899 \times 10^1$ | $1.5608 \times 10^1$ | $9.4278 \times 10^0$ | $1.1527 \times 10^1$ |
| CEC_04 | Mean | $1.9024 \times 10^3$ | $1.9015 \times 10^3$ | $3.9200 \times 10^5$ | $1.9028 \times 10^3$ | $1.9116 \times 10^3$ | $1.9015 \times 10^3$ | $1.9014 \times 10^3$ | $1.9011 \times 10^3$ |
| | Std | $1.5132 \times 10^0$ | $4.7478 \times 10^{-1}$ | $2.0400 \times 10^5$ | $1.1137 \times 10^0$ | $8.0854 \times 10^0$ | $4.6367 \times 10^{-1}$ | $6.4272 \times 10^{-1}$ | $6.8410 \times 10^{-1}$ |
| CEC_05 | Mean | $4.7672 \times 10^3$ | $2.0900 \times 10^4$ | $4.6200 \times 10^5$ | $1.1600 \times 10^5$ | $5.8000 \times 10^5$ | $3.2100 \times 10^4$ | $6.7593 \times 10^3$ | $5.0581 \times 10^3$ |
| | Std | $4.2784 \times 10^3$ | $4.6800 \times 10^4$ | $1.1100 \times 10^5$ | $1.9300 \times 10^5$ | $8.5200 \times 10^5$ | $7.2900 \times 10^4$ | $4.2953 \times 10^3$ | $3.3481 \times 10^3$ |
| CEC_06 | Mean | $1.7312 \times 10^3$ | $1.7684 \times 10^3$ | $2.2222 \times 10^3$ | $1.7776 \times 10^3$ | $1.8431 \times 10^3$ | $1.7573 \times 10^3$ | $1.7587 \times 10^3$ | $1.8632 \times 10^3$ |
| | Std | $1.2285 \times 10^2$ | $9.1820 \times 10^1$ | $2.0670 \times 10^2$ | $1.1171 \times 10^2$ | $1.0223 \times 10^2$ | $8.6052 \times 10^1$ | $1.0387 \times 10^2$ | $1.0503 \times 10^2$ |
| CEC_07 | Mean | $7.0311 \times 10^3$ | $6.5603 \times 10^3$ | $2.9700 \times 10^6$ | $1.8000 \times 10^4$ | $3.4500 \times 10^5$ | $7.3733 \times 10^3$ | $7.5164 \times 10^3$ | $6.0549 \times 10^3$ |
| | Std | $7.7063 \times 10^3$ | $6.4319 \times 10^3$ | $3.8500 \times 10^6$ | $3.7100 \times 10^4$ | $5.3700 \times 10^5$ | $4.9714 \times 10^3$ | $6.1260 \times 10^3$ | $2.6469 \times 10^3$ |
| CEC_08 | Mean | $2.2974 \times 10^3$ | $2.4032 \times 10^3$ | $3.5700 \times 10^3$ | $2.3384 \times 10^3$ | $2.4289 \times 10^3$ | $2.3012 \times 10^3$ | $2.3861 \times 10^3$ | $2.4193 \times 10^3$ |
| | Std | $2.2368 \times 10^1$ | $3.1043 \times 10^1$ | $3.7790 \times 10^2$ | $9.1745 \times 10^1$ | $3.3946 \times 10^1$ | $1.4399 \times 10^1$ | $2.6287 \times 10^2$ | $3.7983 \times 10^2$ |
| CEC_09 | Mean | $2.7018 \times 10^3$ | $2.7599 \times 10^3$ | $2.9038 \times 10^3$ | $2.7449 \times 10^3$ | $2.7752 \times 10^3$ | $2.7330 \times 10^3$ | $2.7514 \times 10^3$ | $2.7922 \times 10^3$ |
| | Std | $1.0302 \times 10^2$ | $1.0211 \times 10^1$ | $9.9242 \times 10^1$ | $4.4570 \times 10^1$ | $6.2479 \times 10^1$ | $6.4081 \times 10^1$ | $9.5351 \times 10^0$ | $1.0509 \times 10^2$ |
| CEC_10 | Mean | $2.9231 \times 10^3$ | $2.9323 \times 10^3$ | $3.6576 \times 10^3$ | $2.9366 \times 10^3$ | $2.9545 \times 10^3$ | $2.9289 \times 10^3$ | $2.9290 \times 10^3$ | $2.9234 \times 10^3$ |
| | Std | $2.2799 \times 10^1$ | $3.1577 \times 10^1$ | $4.0387 \times 10^2$ | $2.4483 \times 10^1$ | $6.9125 \times 10^1$ | $2.4243 \times 10^1$ | $2.9085 \times 10^1$ | $2.3865 \times 10^1$ |
| Average rank | | 2.3 | 3.9 | 7.9 | 4.6 | 6.9 | 3.3 | 3.7 | 3.4 |
| Rank | | 1 | 5 | 8 | 6 | 7 | 2 | 4 | 3 |

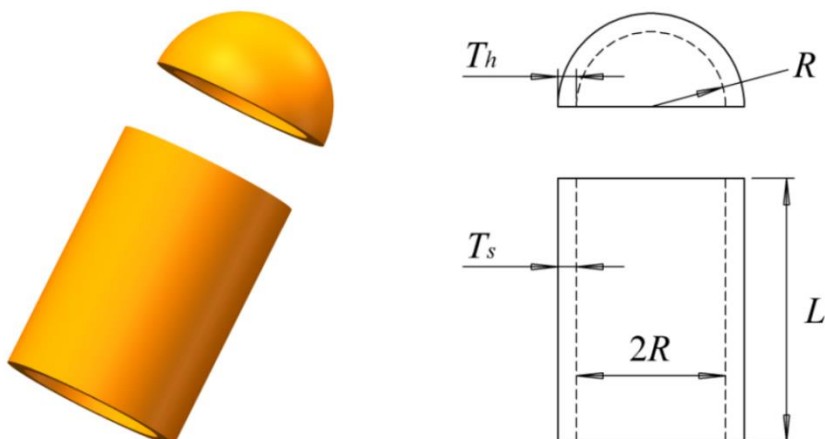

**Figure 5.** Pressure vessel design problem: model diagram (**left**) and structure parameters (**right**).

From Table 13, it can be observed that the DESMAOA was able to obtain the best results in six functions: CEC_02, CEC_05, CEC_06, CEC_08, CEC_09, and CEC_10. Thus, we can find that the DESMAOA has good performance in hybrid and composition test functions. By comparing it with other optimization algorithms, we found that DESMAOA showed very competitive performance for these CEC2021 test functions. Moreover, the Friedman's ranking test was also used to evaluate the performance of DESMAOA. The

average rank and rank were also given in Table 13. It can be seen that DESMAOA obtained the best statistical ranking result among these algorithms.

Therefore, the results of CEC2021 test functions also showed the high performance for solving optimization problems.

## 5. Applicability for Solving Engineering Design Problems

This section reports the three classical engineering design problems we employed to evaluate the capability of DESMAOA to solve practical problems, which were the pressure vessel design problem, three-bar truss design problem, and tension/compression spring design problem. In the same way, 30 search agents and 500 iterations were utilized in the design procedure of engineering problems for a fair comparison. Meanwhile, other related results of optimization algorithms proposed by scholars are also given and compared with proposed algorithm here. Detailed descriptions are shown below.

### 5.1. Pressure Vessel Design

The design of the pressure vessel is an optimization problem with four variables and four constraints in the industrial field [43]. The lowest cost of pressure vessel was the ultimate goal. The structure of pressure vessel is shown in Figure 5. The four design variables were the thickness of the shell ($T_s$), thickness of the head ($T_h$), inner radius ($R$), and length of the cylindrical section ($L$). Table 14 lists the comparison between DESMAOA and other competitor algorithms. From Table 14, we can see that DESMAOA was capable of finding the optimal solution with the lowest cost.

**Table 14.** Optimal results for comparative algorithms on the pressure vessel design problem.

| Algorithm | Optimal Values for Variables | | | | Optimal Cost |
|---|---|---|---|---|---|
| | $T_s$ | $T_h$ | $R$ | $L$ | |
| DESMAOA | $7.943124 \times 10^{-1}$ | $3.927124 \times 10^{-1}$ | $4.288001 \times 10^1$ | $1.671866 \times 10^2$ | $5.8363262 \times 10^3$ |
| SMA [33] | $7.931 \times 10^{-1}$ | $3.932 \times 10^{-1}$ | $4.06711 \times 10^1$ | $1.962178 \times 10^2$ | $5.9941857 \times 10^3$ |
| AOA [34] | $8.303737 \times 10^{-1}$ | $4.162057 \times 10^{-1}$ | $4.275127 \times 10^1$ | $1.693454 \times 10^2$ | $6.0487844 \times 10^3$ |
| MVO [5] | $8.125 \times 10^{-1}$ | $4.375 \times 10^{-1}$ | $4.2090738 \times 10^1$ | $1.7673869 \times 10^2$ | $6.0608066 \times 10^3$ |
| WOA [39] | $8.12500 \times 10^{-1}$ | $4.37500 \times 10^{-1}$ | $4.2098209 \times 10^1$ | $1.76638998 \times 10^2$ | $6.0597410 \times 10^3$ |
| MFO [44] | $8.125 \times 10^{-1}$ | $4.375 \times 10^{-1}$ | $4.2098445 \times 10^1$ | $1.76636596 \times 10^2$ | $6.0597143 \times 10^3$ |
| GWO [38] | $8.125 \times 10^{-1}$ | $4.345 \times 10^{-1}$ | $4.20892 \times 10^1$ | $1.767587 \times 10^2$ | $6.0515639 \times 10^3$ |
| MOSCA [45] | $7.781909 \times 10^{-1}$ | $3.830476 \times 10^{-1}$ | $4.03207539 \times 10^1$ | $1.999841994 \times 10^2$ | $5.88071150 \times 10^3$ |
| LWOA [46] | $7.78858 \times 10^{-1}$ | $3.85321 \times 10^{-1}$ | $4.032609 \times 10^1$ | $2.00 \times 10^2$ | $5.893339 \times 10^3$ |
| IMFO [47] | $7.781948 \times 10^{-1}$ | $3.846621 \times 10^{-1}$ | $4.032097 \times 10^1$ | $1.999812 \times 10^2$ | $5.8853778 \times 10^3$ |

### 5.2. Three-Bar Truss Design

The aim of the three-bar truss design is to achieve the lowest weight of three-bar truss with the constraints of stress, deflection, and buckling, which belongs to the field of civil engineering [48]. In this design problem, two parameters $x_1$ (or A1) and $x_2$ (or A2) were involved, as shown in Figure 6. The solutions obtained by the DESMAOA and other representative algorithms are listed in Table 15. It can be seen that the proposed hybrid method apparently outperformed other approaches. Moreover, 30 repeated tests were also performed to evaluate the robustness of the proposed algorithm. The worst value, mean value, best value, and stand deviation were $2.639079 \times 10^2$, $2.638562 \times 10^2$, $2.638523 \times 10^2$, and $1.0451 \times 10^{-2}$. Hence, the statistical results revealed that the proposed algorithm had very stable and superior performance in solving this design problem.

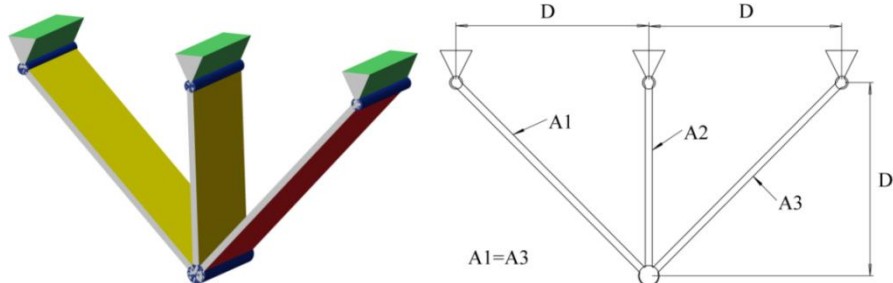

**Figure 6.** Three-bar truss design problem: model diagram (**left**) and structure parameters (**right**).

**Table 15.** Optimal results for comparative algorithms on the three-bar truss design problem.

| Algorithm | Optimal Values for Variables | | Optimal Weight |
|---|---|---|---|
| | $x_1$ | $x_2$ | |
| DESMAOA | $7.882549 \times 10^{-1}$ | $4.085642 \times 10^{-1}$ | $2.638523657 \times 10^2$ |
| SMA [33] | $7.729316 \times 10^{-1}$ | $4.718874 \times 10^{-1}$ | $2.658067955 \times 10^2$ |
| AOA [34] | $7.9369 \times 10^{-1}$ | $3.9426 \times 10^{-1}$ | $2.639154 \times 10^2$ |
| MBA [48] | $7.885650 \times 10^{-1}$ | $4.085597 \times 10^{-1}$ | $2.638958522 \times 10^2$ |
| SSA [40] | $7.88665414 \times 10^{-1}$ | $4.08275784 \times 10^{-1}$ | $2.638958434 \times 10^2$ |
| MFO [44] | $7.88244771 \times 10^{-1}$ | $4.09466906 \times 10^{-1}$ | $2.638959797 \times 10^2$ |
| PSO-DE [49] | $7.886751 \times 10^{-1}$ | $4.082482 \times 10^{-1}$ | $2.638958433 \times 10^2$ |
| HSCAHS [50] | $7.885721 \times 10^{-1}$ | $4.084012 \times 10^{-1}$ | $2.63881992 \times 10^2$ |

*5.3. Tension/Compression Spring Design*

In the design of a tension/compression spring [51], the objective is to obtain the minimum optimal weight under three constraints: (1) shear stress, (2) surge frequency, and (3) deflection. As shown in Figure 7, there were three variables that needed to be considered. They were the wire diameter ($d$), mean coil diameter ($D$), and the number of active coils ($N$). The results of DESMAOA and other comparative algorithms are listed in Table 16. By comparison, the proposed DESMAOA achieved the best solution for this problem, which was $5.44827 \times 10^{-2}$, $4.83109 \times 10^{-1}$, and $5.746128 \times 10^0$ for $d$, $D$, and $N$, respectively. Moreover, the optimal weight was $1.11083 \times 10^{-2}$.

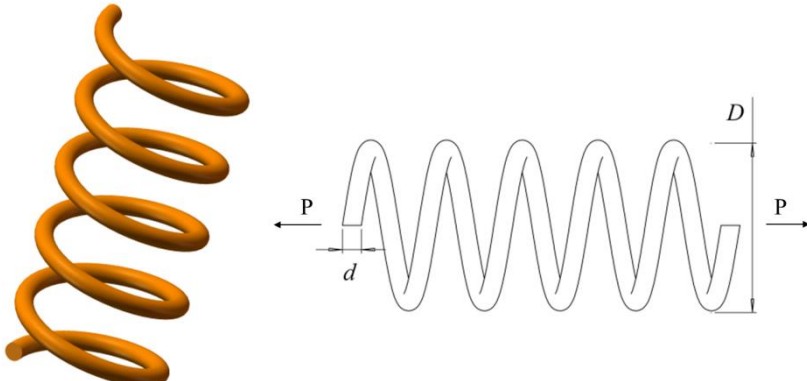

**Figure 7.** Tension/compression spring design problem: model diagram (**left**) and structure parameters (**right**).

**Table 16.** Optimal results for comparative algorithms on the tension/compression spring design problem.

| Algorithm | Optimal Values for Variables | | | Optimal Weight |
|---|---|---|---|---|
| | *d* | *D* | *p* | |
| DESMAOA | $5.44827 \times 10^{-2}$ | $4.83109 \times 10^{-1}$ | $5.746128 \times 10^{0}$ | $1.11083 \times 10^{-2}$ |
| SMA [33] | $5.8992 \times 10^{-2}$ | $6.23402 \times 10^{-1}$ | $3.590304 \times 10^{0}$ | $1.2128 \times 10^{-2}$ |
| AOA [34] | $5.00 \times 10^{-2}$ | $3.49809 \times 10^{-1}$ | $1.18637 \times 10^{1}$ | $1.2124 \times 10^{-2}$ |
| MVO [5] | $5.251 \times 10^{-2}$ | $3.7602 \times 10^{-1}$ | $1.033513 \times 10^{1}$ | $1.2790 \times 10^{-2}$ |
| AO [14] | $5.02439 \times 10^{-2}$ | $3.5262 \times 10^{-1}$ | $1.05425 \times 10^{1}$ | $1.1165 \times 10^{-2}$ |
| SSA [40] | $5.1207 \times 10^{-2}$ | $3.45215 \times 10^{-1}$ | $1.2004032 \times 10^{1}$ | $1.26763 \times 10^{-2}$ |
| GWO [38] | $5.169 \times 10^{-2}$ | $3.56737 \times 10^{-1}$ | $1.128885 \times 10^{1}$ | $1.2666 \times 10^{-2}$ |
| GSA [6] | $5.0276 \times 10^{-2}$ | $3.23680 \times 10^{-1}$ | $1.3525410 \times 10^{1}$ | $1.27022 \times 10^{-2}$ |
| WSA [51] | $5.168626 \times 10^{-2}$ | $3.5665047 \times 10^{-1}$ | $1.129291654 \times 10^{1}$ | $1.267061 \times 10^{-2}$ |

## 6. Conclusions and Future Works

To overcome the shortcomings of basic meta-heuristic algorithms, this paper presents an effective deep ensemble method of two very new optimization algorithms, i.e., the SMA and AOA. A preliminary hybrid of these two algorithms was firstly conducted to enhance the capability of exploration. Then, two strategies were integrated to the hybridized algorithm to assist it to jump out of the local minima and improve the accuracy of the solution. The performance of proposed DESMAOA was extensively analyzed by using 23 classical test functions.

First, different combinations of SMAOA and two strategies were analyzed and discussed. The results revealed the effectiveness of proposed strategies. Then, the results of DESMAOA were compared with SMA, AOA, and five well-known algorithms. The results showed that the proposed method had the advantages of both SMA and AOA and that it also was evidently better than other comparison algorithms. Afterward, experimental tests in high dimensional environments (50, 200, and 1000) were also investigated among these comparative algorithms, and the results of scalability test also confirmed the superior performance of the proposed method. Finally, the proposed DESMAOA was employed to deal with three engineering design problems. The results show that the proposed method was good at solving these problems, and in particular it was very stable when solving the three-bar truss design problem.

As future perspectives, the DESMAOA can be utilized to solve more optimization problems in other disciplines, such as the feature selection, training of multi-layer perceptron neural network, and image processing. Another investigation is to consider the implementation of this hybrid method on other optimization algorithms for better optimization performance.

**Author Contributions:** Conceptualization, R.Z. and H.J.; methodology, R.Z. and H.J.; software, R.Z. and S.W.; validation, R.Z., H.J. and L.A.; formal analysis, R.Z. and S.W.; investigation, R.Z. and H.J.; resources, R.Z., H.J. and L.A.; data curation, R.Z.; writing—original draft preparation, R.Z.; writing—review and editing, R.Z. and H.J.; visualization, Q.L.; supervision, H.J. and L.A.; project administration, R.Z. and H.J.; funding acquisition, R.Z. and H.J. All authors have read and agreed to the published version of the manuscript.

**Funding:** This work was supported by the Sanming University introduces high-level talents to start scientific research funding support project (21YG01, 20YG14), Fujian Natural Science Foundation Project (2021J011128), Guiding science and technology projects in Sanming City (2021-S-8), Educational research projects of young and middle-aged teachers in Fujian Province (JAT200618), Scientific research and development fund of Sanming University (B202009), and Funded By Open Research Fund Program of Fujian Provincial Key Laboratory of Agriculture Internet of Things Application (ZD2101).

**Data Availability Statement:** Not applicable.

**Conflicts of Interest:** The authors declare no conflict of interest.

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
