# Peer review of "Deep Ensemble of Slime Mold Algorithm and Arithmetic Optimization Algorithm for Global Optimization"

_processes, doi:10.3390/pr9101774_

Round 1

Reviewer 1 Report

The authors propose a new hybrid algorithm based on slime mould algorithm (SMA) and arithmetic optimization algorithm (AOA) to improve the optimization capability of original algorithms. First, the slime mould algorithm (SMA) is hybridized by arithmetic optimization algorithm (AOA) to improve the exploration capability of original SMA. Second, it is applied the subtraction and addition strategy from AOA to enhance the exploitation ability of hybrid algorithm. The optimization performance of the proposed new hybrid algorithm is analyzed by using 23 classical benchmark functions, and four typical engineering design problems. The results also indicate that new hybrid algorithm has very promising performance in these problems.

The text needs minor changes, and the results need to be redone for the benchmark functions of the CEC 2021, which will require some time. Several of the 23 benchmark functions used in the article are also listed in CEC 2021.

The article needs the following specific changes:

- Introduction
1) Pag. 4, Line 94: Is DESMAOA the algorithm of items (2) and (3)? If so, it's important to make it clear here.

- Preliminaries
2) Line 113-115: There are some punctuation errors in this sentence excerpt.

3) Pag. 6, Line 146: Why is alpha equal to 5?

- The proposed hybridized algorithm (DESMAOA)
4) Algorithm 3: Pseudo-code of DESMAOA: correct 'Update position Vi1 by Equation (1) (1)'

- Experimental results and discussions
5) Pag. 9, Line 224: The reference [37] is from 2005. For an article from 2021 it is necessary to use the benchmark functions of the CEC2021 ('Problem Definitions and Evaluation Criteria for the CEC 2021 Special Session and Competition on Single Objective Bound Constrained Numerical Optimization', available on http://home.elka.pw.edu.pl/~ewarchul/cec2021-specification.pdf).

6) Pag. 12, Line 291: Why use the Friedman test?

- Applicability for solving engineering design problems
7) Pag. 23, Line 395: put a reference for 'three-bar truss design'.

8) In subsection '5.3. Three-bar truss design' is said "Moreover, thirty times of tests are also performed to evaluate the robustness of proposed algorithm.". Has this been done in the other subsections of section '5. Applicability for solving engineering design problems' ? If so, this information should be placed at the beginning of section '5'.

Author Response

RE: Manuscript processes-1366373

Author:

Rong Zheng    E-mail: zhengr@fjsmu.edu.cn

Rong Zheng    E-mail: zhengr@fjsmu.edu.cn

Heming Jia     E-mail: jiaheming@fjsmu.edu.cn

Response to Reviewer 1:

Question1: Pag. 4, Line 94: Is DESMAOA the algorithm of items (2) and (3)? If so, it's important to make it clear here.

Answer: The description of 4th contribution is modified to make it clear. Please refer to the revised manuscript.

Question2: Line 113-115: There are some punctuation errors in this sentence excerpt.

Answer: The punctuation errors and grammar problems in the first manuscript are carefully revised in the revised manuscript.

Question3: Pag. 6, Line 146: Why is alpha equal to 5?

Answer: The alpha is a constant value set in AOA. The AOA has better performance when alpha is equal to 5.

Question4: Algorithm 3: Pseudo-code of DESMAOA: correct 'Update position Vi1 by Equation (1) (1)'

Answer: The “Update position Vi1 by Equation (1) (1)” is modified by “Update position Vi1 by Equation (1) (1)'”.

Question5: Pag. 9, Line 224: The reference [37] is from 2005. For an article from 2021 it is necessary to use the benchmark functions of the CEC2021 ('Problem Definitions and Evaluation Criteria for the CEC 2021 Special Session and Competition on Single Objective Bound Constrained Numerical Optimization', available on http://home.elka.pw.edu.pl/~ewarchul/cec2021-specification.pdf).

Answer: In the revised draft, we added the experimental results of IEEE CEC2021 standard test functions in Section 4.3 to further illustrate the performance of proposed DESMAOA. For detailed information, please refer to Section 4.3.

Question6: Pag. 12, Line 291: Why use the Friedman test?

Answer: The result of Friedman test can reveal the overall performance ranking of the comparative algorithms to the test functions.

Question7: Pag. 23, Line 395: put a reference for 'three-bar truss design'.

Answer: A related reference has been added for the 'three-bar truss design'.

Question8: In subsection '5.3. Three-bar truss design' is said "Moreover, thirty times of tests are also performed to evaluate the robustness of proposed algorithm.". Has this been done in the other subsections of section '5. Applicability for solving engineering design problems' ? If so, this information should be placed at the beginning of section '5'.

Answer: Thirty times of tests are also performed for the speed reducer design problem because we find that the proposed algorithm also has stable performance on solving this problem. However, for other two engineering design problems discussed in this manuscript, the performance of proposed algorithm is not so stable. Thus the thirty times of tests are not conducted for the left two engineering design problems.

At last, Many thanks to the reviewers for their valuable comments.

Reviewer 2 Report

The paper proposes a hybrid population algorithm. It is worth mentioning that there are a wide variety of algorithms that are meant to solve a particular group of problems due to the "no free lunch" theorem. This fact is mentioned in the paper. However, often the NFT theorem is wrongly interpreted. The current paper is also the case. The NFT theorem states that certain initial guesses cannot be significantly improved with the given optimization algorithm for a given class of problems. Only one benchmark with several cases is present in the paper, and the algorithm's applicability conclusions are not drawn.

 To start with, the idea of combining the global and local search is not novel. Some papers on this topic describe a combination of local gradient descent and global population search. Others describe the varying exploration vs. exploitation ratio of one algorithm. The current paper describes a combination of two algorithms - one with higher exploration ability and the other with higher exploitation ability. The choice of strategy is governed by a varying probability to choose exploitation more often during the population evolution process. 

Such an idea by itself may be the topic of the paper. However, in the current state, it is presented not sufficiently clear. I will put some major comments section-by-section

Section 1. Apart from typos such as "gradient decent", it contains a somewhat inappropriate division of the algorithms to "exact" and "approximate" ones. Whereas it seems reasonable, from the scheme Fig. 1, we extract that the "approximate" class is equal to the heuristic algorithms. Further in the text, we understand that we are within the numerical optimization domain, and we cannot speak about the "exact" methods overall. I think such division, as well as any other, is not necessary here and does not support any ideas of the paper. Thus, the introductory paragraphs are to be reconsidered.

The following paragraphs show plenty of meta-heuristic algorithms with the conclusion "In this paper, the hybridized algorithm based on the SMA and AOA is proposed to solve global optimization problems. Both algorithms have the merits of simplicity, efficiency and flexibility, and also are easy to implement. ". It is not enough to choose the particular algorithms for hybridisation. I believe that the authors should have more a priori grounding, which could be stated explicitly in the introduction.

Section 2. I think Eq. 3 contains a typo, and instead of "condition" and "others", explicit criteria should be placed. 

Section 2.2 Has inappropriate reference "Arithmetic optimization algorithm (AOA) is a very new meta-heuristic method proposed by Jordan scholar Abualigah and others in 2021 [35]." Apart from the Jordan scholar Abualigah being the only full name stated in the paper, which somewhat discriminates the other researchers, other references do not have author names and/or year of publication. In case the authors immensely respect the researcher, there is another place to express it in the paper, such as the Acknowledgments section. 

Eq. 7 contains the ÷ sign, which in some literature is used as the integer division sign. If it is not the case, it should be replaced with more common / or explained if it is the case. Also, crosses for multiplication are commonly omitted if the operation is not ambiguous. Otherwise, it is more common to see a dot since the cross is more often is referred to as cross-product.

Section 3. Describes the additions done to the AOA and SMA to make DESMAOA. It is again done without any preliminary assessments.

Section 4. It contains benchmarking results. Major comment - benchmarks are not used to compare algorithms performance. The guide to correct benchmarks using may be found in [1]. The idea is that a given optimization algorithm may solve one class of problems better than another, which is the directly NFT theorem content. The primary purpose of the benchmark is that every correct algorithm should be able to optimize most of the functions with a sufficiently large number of steps. Those functions that were not optimized are the class that cannot be solved using the particular algorithm.

[1] Derrac, J., García, S., Molina, D., & Herrera, F. (2011). A practical tutorial on the use of nonparametric statistical tests as a methodology for comparing evolutionary and swarm intelligence algorithms. Swarm and Evolutionary Computation1(1), 3-18.

Particularly, Table 3 contains "0" and "5.63E−12". Why the latter is not "0"? What exactly "0" is?

Table 4 shows hyperparameter values. There are no words on how the DESMAOA hyperparameter set was chosen and why this particular one. If it was tuned on the benchmark, then all algorithms are in "unequal" positions.

Table 5 contains "0" and "5.4347E−323". Again, why latter is not "0"?

Table 6 contains p-values to assess the significance of the DESMAOA quality dominance. The values are the same in most cases, which casts doubt on an experiment setup as a whole.

Overall the numerical results in this section are not correctly present and interpreted. The excessive number of decimal digits further increases the misunderstanding.

Overall, Section 4 makes an impression that a clear and correct setup and controls are missing in the benchmark experiment, and a more rigorous description of decisions to choose such an experimental setup is required.

Section 5 seems to be the most valuable in the paper. The real problem shows that DESMAOA is, for several cases, insignificantly better than the other algorithms. However, for most of the experiments, it is evidently showing the same or worse performance. Again, the threat of the numerical results cast doubt on the obtained results overall.

Section 6 is not agreeing with the results of the previous sections. Again, all the experiments have incorrect setup and numerical analysis.

 I cannot advise the paper to be published in its current form. Significant reconsideration of content, experimental setups, and analysis is required. The changes required rather make a new paper than the significant revision of the current paper. As a target paper, which has a correct conclusions and analysis [2] may be proposed.

[2] Vesterstrom, J., & Thomsen, R. (2004, June). A comparative study of differential evolution, particle swarm optimization, and evolutionary algorithms on numerical benchmark problems. In Proceedings of the 2004 congress on evolutionary computation (IEEE Cat. No. 04TH8753) (Vol. 2, pp. 1980-1987). IEEE.

Author Response

RE: Manuscript processes-1366373

Author:

Rong Zheng    E-mail: zhengr@fjsmu.edu.cn

Rong Zheng    E-mail: zhengr@fjsmu.edu.cn

Heming Jia     E-mail: jiaheming@fjsmu.edu.cn

Response to Reviewer 2:

Question1: Section 1. Apart from typos such as "gradient decent", it contains a somewhat inappropriate division of the algorithms to "exact" and "approximate" ones. Whereas it seems reasonable, from the scheme Fig. 1, we extract that the "approximate" class is equal to the heuristic algorithms. Further in the text, we understand that we are within the numerical optimization domain, and we cannot speak about the "exact" methods overall. I think such division, as well as any other, is not necessary here and does not support any ideas of the paper. Thus, the introductory paragraphs are to be reconsidered.

Answer: The typos such as "gradient decent" in the manuscript have been carefully corrected. To make the introduction and the main idea of this paper clear, we have deleted the Fig. 1. And also the introductory paragraphs are carefully modified.

Question2: The following paragraphs show plenty of meta-heuristic algorithms with the conclusion "In this paper, the hybridized algorithm based on the SMA and AOA is proposed to solve global optimization problems. Both algorithms have the merits of simplicity, efficiency and flexibility, and also are easy to implement. ". It is not enough to choose the particular algorithms for hybridization. I believe that the authors should have more a priori grounding, which could be stated explicitly in the introduction.

Answer: A supplementary explanation for hybridizing SMA and AOA are described in the introduction part. Specifically, “The SMA has good population diversity and stable performance when solving optimization problems. However, it gets stuck in local optima sometimes for the limited global search capability. On the contrary, the AOA has powerful exploration capability by using the arithmetic operators. But the performance of AOA is not stable because of the poor population diversity. Therefore, the SMA and AOA) are considered to be hybridized together in this paper for solving the global optimization problems. ”

Question3: Section 2. I think Eq. 3 contains a typo, and instead of "condition" and "others", explicit criteria should be placed.

Answer: The explicit criteria is given in Eq. 3 to replace the "condition" and "others".

Question4: Section 2.2 Has inappropriate reference "Arithmetic optimization algorithm (AOA) is a very new meta-heuristic method proposed by Jordan scholar Abualigah and others in 2021 [35]." Apart from the Jordan scholar Abualigah being the only full name stated in the paper, which somewhat discriminates the other researchers, other references do not have author names and/or year of publication. In case the authors immensely respect the researcher, there is another place to express it in the paper, such as the Acknowledgments section.

Answer: In the revised draft, the relevant expression has been amended. The authors of SMA also have been mentioned.

Question5: Eq. 7 contains the ÷ sign, which in some literature is used as the integer division sign. If it is not the case, it should be replaced with more common / or explained if it is the case. Also, crosses for multiplication are commonly omitted if the operation is not ambiguous. Otherwise, it is more common to see a dot since the cross is more often is referred to as cross-product.

Answer: The signs in the expression have been modified appropriately.

Question6: Section 3. Describes the additions done to the AOA and SMA to make DESMAOA. It is again done without any preliminary assessments.

Answer: A supplementary explanation for hybridizing SMA and AOA are described in the introduction part. And also in the Section 3, the characteristics of the two algorithms are described firstly.

Question7: Section 4. It contains benchmarking results. Major comment - benchmarks are not used to compare algorithms performance. The guide to correct benchmarks using may be found in [1]. The idea is that a given optimization algorithm may solve one class of problems better than another, which is the directly NFT theorem content. The primary purpose of the benchmark is that every correct algorithm should be able to optimize most of the functions with a sufficiently large number of steps. Those functions that were not optimized are the class that cannot be solved using the particular algorithm.

[1] Derrac, J., García, S., Molina, D., & Herrera, F. (2011). A practical tutorial on the use of nonparametric statistical tests as a methodology for comparing evolutionary and swarm intelligence algorithms. Swarm and Evolutionary Computation, 1(1), 3-18.

Answer: Lots of thanks for this comment. I have read the reference provided by the reviewer. And the related expressions in the manuscript have been modified.

Question8: Particularly, Table 3 contains "0" and "5.63E−12". Why the latter is not "0"? What exactly "0" is?

Answer: In the experiments, we set the number of iterations and population size as 500 and 30 for all optimization algorithms. Thus, the optimization algorithms may or may not find the theoretical optimization values within limited sources. For a good optimization algorithm, the obtained fitness results can equal to the exact theoretical optimization values (such as 0, or other values). And the optimization algorithm also will show very stable performance, which makes the standard deviation values be zero. On the contrary, for a not so good optimization algorithm, the obtained results may close to the theoretical optimization values. And also the algorithm will show unstable performance, which makes the standard deviation values be a positive (such as 5.63E−12, though it is very small.). Similar results can be referred to the references [1] and [2]

[1] Li, S.; Chen, H.; Wang, M.; Heidari, A.A.; Mirjalili; S. Slime Mould Algorithm: A new method for stochastic optimization. Future Gener. Comput. Syst. 2020, 111, 300–323.

[2] Abdollahzadeh, B.; Gharehchopogh, F.S.; Mirjalili, S. African Vultures Optimization Algorithm: A New Nature-Inspired Metaheuristic Algorithm for Global Optimization Problems. Comput. Ind. Eng. 2021, 1.

Question9: Table 4 shows hyperparameter values. There are no words on how the DESMAOA hyperparameter set was chosen and why this particular one. If it was tuned on the benchmark, then all algorithms are in "unequal" positions.

Answer: We have added the description to explain the setting of parameter values for the DESMAOA. The related hyperparameter values of DESMAOA are same as those used in original algorithms without any other adjustments. Moreover, we also conduct the tests of CEC2021 to further investigate the DESMAOA’s performance.

Question10: Table 5 contains "0" and "5.4347E−323". Again, why latter is not "0"?

Answer: Please refer to the answer of Question8.

Question11: Table 6 contains p-values to assess the significance of the DESMAOA quality dominance. The values are the same in most cases, which casts doubt on an experiment setup as a whole.

Answer: To calculate the p-values between two optimization algorithms for a test function, 15 independent tests are conducted for each algorithm. Thus according to the definition of the Wilcoxon signed-rank test, for a particular case, one algorithm always gets better results than those of the other algorithm, the result of p-value should be 6.10E-05. In our experiments, this situation happens often because the proposed DESMAOA can achieve better fitness that the comparative algorithms. Similar results can also be found in previous references [1] and [2].

Question12: Overall the numerical results in this section are not correctly present and interpreted. The excessive number of decimal digits further increases the misunderstanding. Overall, Section 4 makes an impression that a clear and correct setup and controls are missing in the benchmark experiment, and a more rigorous description of decisions to choose such an experimental setup is required.

Answer: To present the numerical results more clearly, these results in Section 4 and Section 5 have been modified by the scientific notation. And relevant expressions have also been amended accordingly.

Question13: Section 5 seems to be the most valuable in the paper. The real problem shows that DESMAOA is, for several cases, insignificantly better than the other algorithms. However, for most of the experiments, it is evidently showing the same or worse performance. Again, the threat of the numerical results cast doubt on the obtained results overall.

Answer: The tests of engineering design problems are used to evaluate whether the  proposed algorithm can deal with the practical problems. By comparing the obtained results of DESMAOA and other comparative algorithms, DESMAOA can obtain the best results for Pressure vessel design and Three-bar truss design problems and comparative results for Welded beam design and Speed reducer design problems. Thus in order to fully illustrate the performance of DESMAOA, the results of Pressure vessel design and Three-bar truss design problems are selected in the revised manuscript.

Question14: Section 6 is not agreeing with the results of the previous sections. Again, all the experiments have incorrect setup and numerical analysis.

Answer: The first draft was carefully revised based on valuable comments from reviewers. Expressions have been modified to be more accurate. We have added the results of CEC2021 test functions in Section 4.3. The results of engineering design problems are also modified.

At last, Many thanks to the reviewers for their valuable comments.

Round 2

Reviewer 2 Report

This is my second review of the paper.

The current version is somewhat improved, yet I doubt that the recent quality improvement is sufficient to publish the paper.

My main concern is the numerical results treatment. In the numerical optimization algorithms domain, such are considered in the paper. We cannot tell about the "exact" values. However, for some reason, authors continue to use such terminology, which is confusing. Therefore I am afraid I fundamentally have to disagree with the first paragraph of Section 1. See also some reasoning for this below.

Overall, Sections 1-3 are improved. However, further numerical treatment in section 4 and answering Question 8 cast further doubt that the numerical experiments were done and assessed correctly. 

From Table 3, we can understand that in some cases, the results either have a confidence interval around 200% of mean value in others around 0%. In particular, F10 in table 3 shows the same unexplained value "8.8818E−16" for four different cases, which is very strange for the population algorithm. This casts doubt on how the initial population is formed (which is not described well in the paper). All these facts make the reader consider all results in Table 3 as unrealistic.

Table 4 contain values with exponents of order -100. On the one hand, this is unrealistic to consider practically due to the insufficient quality of measuring devices. On the other hand, in numerical analysis, it is considered as "0" or "~0" since zero is not a constant in the numerical analysis domain, unlike symbolic computation.

Tables 6 and 9-11 show that the p-value test is not adequate to prove the null hypothesis and cast doubt on the correctness of the comparison results. The notes on how to use criteria in statistics may be found in [1]

[1] Jung I. (2017). Some Facts That You Might Be Unaware of About the P-Value. Archives of plastic surgery44(2), 93–94. https://doi.org/10.5999/aps.2017.44.2.93

The conclusion for Section 4 is not done by the authors implicitly. Nevertheless, we may conclude from Tables 5-11 that DESMAOA is better for almost every task. It may be true and, in the case of correct numerical treatment, shows that a combination of global and local search is better than any single algorithm. That implies that the comparison is not done correctly and should include other combinations as well.

The solution to remove cases where DESMAOA is not outperforming in Section 5 is strange. Numerical results are still presented with an excessive amount of decimal digits.

To conclude, I am still cannot recommend the paper for publication and think that more thorough reconsideration of the paper content is required. Most of the experiment results are not convincing and coherent. The conclusions are not correctly drawn. I advise the authors to take more time than the one week that MPDI usually recommends and make complete reconsideration and analysis of the experimental results.

Author Response

RE: Manuscript processes-1366373

Author:

Rong Zheng    E-mail: zhengr@fjsmu.edu.cn

Rong Zheng    E-mail: zhengr@fjsmu.edu.cn

Heming Jia     E-mail: jiaheming@fjsmu.edu.cn

Response to Reviewer 2:

Question1: My main concern is the numerical results treatment. In the numerical optimization algorithms domain, such are considered in the paper. We cannot tell about the "exact" values. However, for some reason, authors continue to use such terminology, which is confusing. Therefore I am afraid I fundamentally have to disagree with the first paragraph of Section 1.

Answer: We have understood that the improper use of "exact". In the second revised manuscript, we have reorganized the first paragraph in Section 1, and modified the expression about the traditional and stochastic algorithms.

Question2: Overall, Sections 1-3 are improved. However, further numerical treatment in section 4 and answering Question 8 cast further doubt that the numerical experiments were done and assessed correctly. From Table 3, we can understand that in some cases, the results either have a confidence interval around 200% of mean value in others around 0%. In particular, F10 in table 3 shows the same unexplained value "8.8818E−16" for four different cases, which is very strange for the population algorithm. This casts doubt on how the initial population is formed (which is not described well in the paper). All these facts make the reader consider all results in Table 3 as unrealistic.

Answer: The theoretical optimization value of F10 is 0. However, due to the limited performance of the algorithm, the obtained results of these algorithms only can achieve the same value "8.8818E−16" though many times of testing. This is also reasonable for a stable optimization algorithm and similar results can be referred to the references [1]. On the contrary, the results may have widely divergent when different algorithms deal with different test functions. Thus the results with a confidence interval around 200% or 0% of mean value are also possible. The initial positions of the population are randomly generated, which is also described in the second revised manuscript, i.e. “It is known that the search agents are first randomly generated within the search spaces.”.

[1] Abdollahzadeh, B.; Gharehchopogh, F.S.; Mirjalili, S. African Vultures Optimization Algorithm: A New Nature-Inspired Metaheuristic Algorithm for Global Optimization Problems. Comput. Ind. Eng. 2021, 1.

Question3: Table 4 contain values with exponents of order -100. On the one hand, this is unrealistic to consider practically due to the insufficient quality of measuring devices. On the other hand, in numerical analysis, it is considered as "0" or "~0" since zero is not a constant in the numerical analysis domain, unlike symbolic computation.

Answer: It is right that we cannot obtain the values with exponents of order -100 with sufficient quality of measuring devices. Nevertheless, in our works, we can obtain the high precision results for the standard benchmark functions, which are designed specifically for the optimization problems. And it is meaningful to express like this to fully present the superiority of the algorithm.

Question4: Tables 6 and 9-11 show that the p-value test is not adequate to prove the null hypothesis and cast doubt on the correctness of the comparison results. The notes on how to use criteria in statistics may be found in [1]

[1] Jung I. (2017). Some Facts That You Might Be Unaware of About the P-Value. Archives of plastic surgery, 44(2), 93–94. https://doi.org/10.5999/aps.2017.44.2.93

Answer: According to the Tables 6 and 9-11, the proposed DESMAOA has significant differences compared with the comparative algorithms. By referring to the theoretical optimization results shown in Tables 5 and 7-8, the DESMAOA outperforms other algorithms. Moreover, the results of Friedman ranking test in Table 12 also show the superiority of DESMAOA.

Question5: The conclusion for Section 4 is not done by the authors implicitly. Nevertheless, we may conclude from Tables 5-11 that DESMAOA is better for almost every task. It may be true and, in the case of correct numerical treatment, shows that a combination of global and local search is better than any single algorithm. That implies that the comparison is not done correctly and should include other combinations as well.

Answer: In Section 4, we first investigate the impacts of components. Four kinds of improved algorithms are obtained based on the SMA and AOA: SMAOA, SMAOA combined with RCS, SMAOA combined with SAS, and SMAOA combined with RCS and SAS. According to the test results, we find that the last one has the best performance, which is then used comparison testing with other algorithms. Also, we have conducted other combinations of SMA and AOA and finally we get the best one, i.e. the DESMAOA.

Question6: The solution to remove cases where DESMAOA is not outperforming in Section 5 is strange. Numerical results are still presented with an excessive amount of decimal digits.

Answer: In viewing of that the DESMAOA has shown the same or worse performance on the the cases which are removed, we conduct further tests on other real-world problems. In the second revised manuscript, we shown that the proposed DESMAOA can obtain the best result when solving the tension/compression spring design problem compared with other algorithms. In addition, we also have reasonably modified the presentation of the data, such as the data values are presented in exponential form in Section 4 and decimal form in Section 5.

At last, Many thanks to the reviewer 2 for these careful reviews of our works.

Round 3

Reviewer 2 Report

At the third round of review, I still cannot accept the authors' position concerning the numerical results treatment.

However, I do not want to continue this discussion any further. I would like to advise the authors not to use "0" and exponents of order "E-100" separately for future work to avoid misunderstanding. Usually, they are replaced with the "0" with some description like "values of order lower than E-12 (as an example) are considered as 0" and reduce the number of decimal digits in the results they do not have meaning neither theoretically nor practically.

I would like the authors to reduce the number of decimal digits in all tables before publications.

Author Response

RE: Manuscript processes-1366373

Author:

Rong Zheng    E-mail: zhengr@fjsmu.edu.cn

Rong Zheng    E-mail: zhengr@fjsmu.edu.cn

Heming Jia     E-mail: jiaheming@fjsmu.edu.cn

Response to Reviewer 2:

Question: I would like to advise the authors not to use "0" and exponents of order "E-100" separately for future work to avoid misunderstanding. Usually, they are replaced with the "0" with some description like "values of order lower than E-12 (as an example) are considered as 0" and reduce the number of decimal digits in the results they do not have meaning neither theoretically nor practically. I would like the authors to reduce the number of decimal digits in all tables before publications.

Answer: To make the data representation be clearer, we further modified the results of F8 in Table 8, reducing the number of decimal digits. And some other places also have been modified like this. On the other hand, it is also necessary to keep a certain number of decimal digits for the effective discrimination between two algorithms. We will pay careful attention to the presentation of numerical results in the future work. At last, special thanks to the reviewer for these valuable reviews on our works.
